# Psychological booster shots targeting memory increase long-term resistance against misinformation

Rakoen Maertens [1] ✉, Jon Roozenbeek [2,3], Jon S. Simons [2], Stephan Lewandowsky [4,5], Vanessa Maturo [6], Beth Goldberg [6], Rachel Xu [6] & Sander van der Linden [2]

An increasing number of real-world interventions aim to preemptively protect or inoculate people against misinformation. Inoculation research has demonstrated positive effects on misinformation resilience when measured immediately after treatment via messages, games, or videos. However, very little is currently known about their long-term effectiveness and the mechanisms by which such treatment effects decay over time. We start by proposing three possible models on the mechanisms driving resistance to misinformation. We then report five pre-registered longitudinal experiments ($N_{total} = 11,759$) that investigate the effectiveness of psychological inoculation interventions over time as well as their underlying mechanisms. We find that text-based and video-based inoculation interventions can remain effective for one month—whereas game-based interventions appear to decay more rapidly—and that memory-enhancing booster interventions can enhance the diminishing effects of counter-misinformation interventions. Finally, we propose an integrated memory-motivation model, concluding that misinformation researchers would benefit from integrating knowledge from the cognitive science of memory to design better psychological interventions that can counter misinformation durably over time and at-scale.

Misinformation is a threat to society and the functioning of democracies worldwide[1,2]. It is shown to have impacted a wide variety of critical issues such as vaccine uptake[3–5], support for mitigation of anthropogenic global warming[6–8], and political elections[9,10]. Furthermore, misinformation has also been linked to real-world violence, such as mob violence in India and the burning of 5G installations[11,12].

Many current methods to counter misinformation involve debunking[13]. Such post-hoc corrections can be effective, but a growing body of evidence highlights the advantages of preventing the spread of misinformation proactively[14–16]. One such preemptive approach is psychological inoculation—interventions that warn people about

upcoming misinformation threats (the forewarning) and, using weakened (micro-)doses of misinformation, teach people the skills required to counter-argue and detect the flawed reasoning that underlies misinformation[17–19]. In the past several years, researchers have successfully tested text-based[7,8,20], gamified[21–23], and video-based inoculation interventions[24,25]. Many inoculation interventions focus on specific issues or misleading narratives[7,8,20]. However, inoculation interventions can also provide a more scalable approach to countering misinformation by targeting the underlying rhetorical technique used to manipulate (e.g., using emotional language in a misleading way, increasing polarization, and the use of logical fallacies)[14,26]. In addition,

[1]Department of Experimental Psychology, University of Oxford, Oxford, UK. [2]Department of Psychology, University of Cambridge, Cambridge, UK. [3]Department of War Studies, King's College London, London, UK. [4]School of Psychological Science, University of Bristol, Bristol, UK. [5]University of Potsdam, Potsdam, Germany. [6]Jigsaw, Google LLC, New York, NY, USA. ✉e-mail: rakoen.maertens@psy.ox.ac.uk

even if the participant has already been exposed to the misinformation before the inoculation intervention, the intervention can still be effective in a therapeutic sense[27]. Finally, interventions can be either passive or active[19,27], depending on whether participants have to actively engage with and generate the inoculation content, or passively consume it.

Classical inoculation theory[19] proposes that an inoculation intervention works by (1) increasing the perceived feelings of threat of being influenced by misinformation, which leads to an increased motivation to defend oneself against it, and (2) making people more familiar with the misleading tactics the manipulator could use; taken together, these processes increase people's willingness and ability to resist and counter-argue misinformation[17]. In contrast, some scholars have argued that unlike the threat-motivation view, inoculation effects could also be explained in terms of memory processes such as associative learning and forgetting, and that the effectiveness of inoculation interventions could be determined by memory strength rather than motivation or threat[28,29], or in conjunction with motivation and threat[28,29].

Despite the recent success of inoculation interventions, three crucial insights are still missing in the literature: how long do the effects of inoculation last, what drives the diminishing effects, and how can we maintain the effectiveness over time? The real-world potential of inoculation interventions has been hampered by these knowledge limitations and the questions regarding the mechanisms by which treatment effects dissipate over time remained unanswered. To date, research has not tested long-term effectiveness systematically across different formats of inoculation or directly explored the underlying cognitive mechanisms[30].

In this work, we pursue three research goals: (1) to explore and identify the decay rate of text-based, video-based, and game-based inoculation interventions, (2) to propose a general theory that can account for the underlying mechanisms responsible for effect retention, and (3) to test interventions that can boost the longevity of inoculation effects by targeting memory and motivation. For each intervention we investigate the long-term effectiveness immediately after the inoculation intervention (T0), -10 days after the intervention (T10), and -30 days after the intervention (T30). Participants in a booster group also receive an inoculation booster intervention at T10.

The first intervention (used in Study 1) is a passive, issue-focused, text-based intervention that inoculates participants against misinformation about the scientific consensus on anthropogenic global warming[7,8]. The second intervention (used in Study 2) is an active, technique-focused, online inoculation game (Bad News), in which participants have to create and spread their own misinformation, albeit in severely weakened form (i.e., using humorous examples that highlight the flaws in the misinformation in a safe, controlled environment), as part of a simulated social media environment[22,28]. The third intervention (Studies 3–5) is a short video that inoculates people against a technique that is often used to mislead people. This video-based intervention was shown to be effective at improving people's ability to detect misleading headlines that use a range of different manipulation strategies in a field study on YouTube. The videos have been shown to over 5 million YouTube users as an educational advertisement in the United States[25], with effects lasting up to at least 24 h. Similar videos have been successfully used in Central and Eastern Europe[31], as well as in Germany, with over 42 million views[32]. Yet, despite its wide-scale implementation on social media, its efficacy beyond 24 h remains unknown. For this study, we focus on the long-term effectiveness of just one of the misleading techniques previously researched, namely using emotional language to misguide people. See Fig. 1 for an overview of the different studies, their interventions, and their experimental design.

To disambiguate the underlying mechanisms of the inoculation effects and their longevity, we administer questions to measure (1) motivation and threat, and (2) memory of the refutation, drawing upon the cognitive science literature. See Fig. 2 for a graphical comparison of the theoretical models, and Methods for more details. We also present an integrated account (the memory-motivation model of inoculation), which states that motivational processes are important to improve memory, but that memory is the main predictor for the longevity of inoculation effects. Finally, we note that when we refer to *decay*, we use a purely functional definition of the term: a decrease in effect over time. We do not refer to decay as a possible explanation of forgetting. The results of the study indicate that text-based, video-based, and gamified inoculation interventions all show consistent effects, but with varying longevity. This work also shows that the longevity of the intervention effects is best predicted by how much people remember from the intervention, and to a minor extent motivation, and that researchers and practitioners can develop ways to increase the long-term effectiveness of interventions by focusing on boosting memory.

## Results

### Study 1: text-based inoculation

In Study 1, participants were exposed to misinformation concerning the scientific consensus on anthropogenic global warming 0, -10, or -30 days after reading an inoculation message or completing a control task[8]. We hypothesized that we would replicate the finding that exposure to misinformation (a misleading petition that suggests scientists disagree on human-caused global warming) reduced the reader's *perceived scientific consensus* (PSC, i.e., perceived agreement amongst scientific experts on a 0–100% scale) of anthropogenic global warming (H1) and that an inoculation message can prevent a decrease in PSC (H2). The delay intervals of 10 days (T10) and 30 days (T30) were chosen as we know from Maertens et al.[8] that there is no significant decay of the inoculation intervention after one week, but some scholars suggests that decay can be detected as soon as two weeks after the intervention[30,33]. This timeframe allowed us to test the limits of the effect with the hypothesis that the effect may still be intact after 10 days (H3), but not after 30 days (T30; H4). Our design also allowed us to test the effectiveness of a booster intervention in the form of a repetition of the original intervention at T10, which we expect to top up the effect and reduce its decay at T30, which is 30 days after T0 or 20 days after the booster at T10 (H5). Finally, we tested three hypotheses as to whether both memory and threat-induced motivation are viable predictors of the outcome of inoculation interventions, with inoculation booster interventions expected to improve memory (H6) and motivation (H7) and the inoculation effect expected to be mediated by memory and motivation (H8). See Methods for details on how memory is measured. Supplementary Table 1 presents an overview of the preregistered hypotheses and what evidence was found to support them. To test our hypotheses, we used both frequentist analysis (as preregistered) and Bayesian analyses throughout the paper, showing by and large the same results. We will therefore focus mainly on the preregistered frequentist analyses unless there are clear deviations.

We started by testing [H1] the main effect of the misinformation message and [H2] the main effect of the inoculation message. We found that, in line with our hypotheses, the misinformation message had a negative effect on the perceived scientific consensus (PSC), [H1] $M_{pre} = 84.33$, $M_{post} = 79.53$, $M_{diff} = -4.80$, 95% CI [−7.10, −2.50], $t(301) = -4.11$, $p < 0.001$, $d = -0.237$, 95% CI [−0.351, −0.122], Bayesian posterior mean = −4.72, 95% CI [−7.08, −2.39], $BF_{10} = 224.157$ (error <0.001%), while when an inoculation message was shown before the misinformation message, there was no negative effect but there was a positive effect on the perceived scientific consensus that climate change is human-caused, [H2] $M_{pre} = 84.72$, $M_{post} = 92.06$, $M_{diff} = 7.34$, 95% CI [5.59, 9.10], $t(316) = 8.24$, $p < 0.001$, $d = 0.463$, 95% CI [0.347, 0.578], Bayesian posterior mean = 7.28, 95% CI [5.54, 9.05], $BF_{10} = 1.127e + 12$ (error <0.001%). Although not preregistered, we also

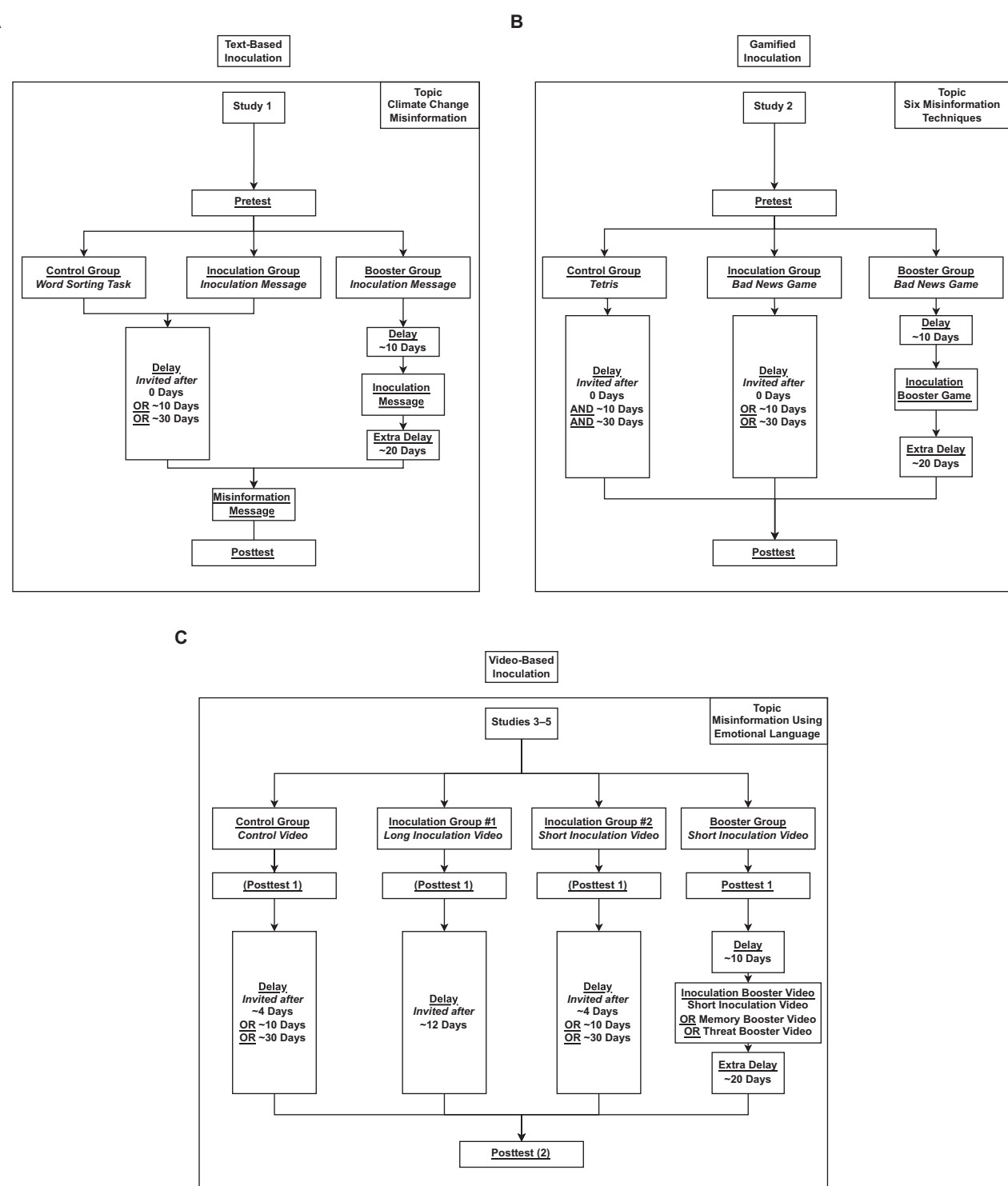

**Fig. 1 | Experimental design flowcharts for studies 1–5.** Flowcharts of the procedure participants went through in text-based Study 1 (**A** topic: climate change), gamified Study 2 (**B** topic: six often used misinformation techniques), and video-based Studies 3–5 (**C** topic: misinformation using emotional language). Each rectangle represents a stage of the study. The arrows depict the order of procedure for participants. Branching means participants were randomly allocated to a path. Pretests and posttests refer to the measurement of inoculation effects.

established the baseline inoculation effect using an ANCOVA to compare the inoculation group with the control group with the pretest core as a covariate, and found the result to be significant with medium-to-large effects, $t(616) = 9.19$, $p_{tukey} < 0.001$, $d = 0.739$, 95% CI [0.576, 0.902], Bayesian posterior mean = 85.80, 95% CI [84.48, 87.11], BF10 = 6.982e + 15 (error = 0.779%).

Having replicated the main effects of interest using within-group analyses, we next investigated the retention of the inoculation effect at T10 ($Mdn = 8$ days) and T30 ($Mdn = 29$ days) using between-group ANCOVA contrasts (comparing the inoculation groups with the control groups) with pretest as a covariate. We first found that the inoculation effect was still significant at 8 days, [H3] $F(1, 514) = 18.94$, $p < 0.001$,

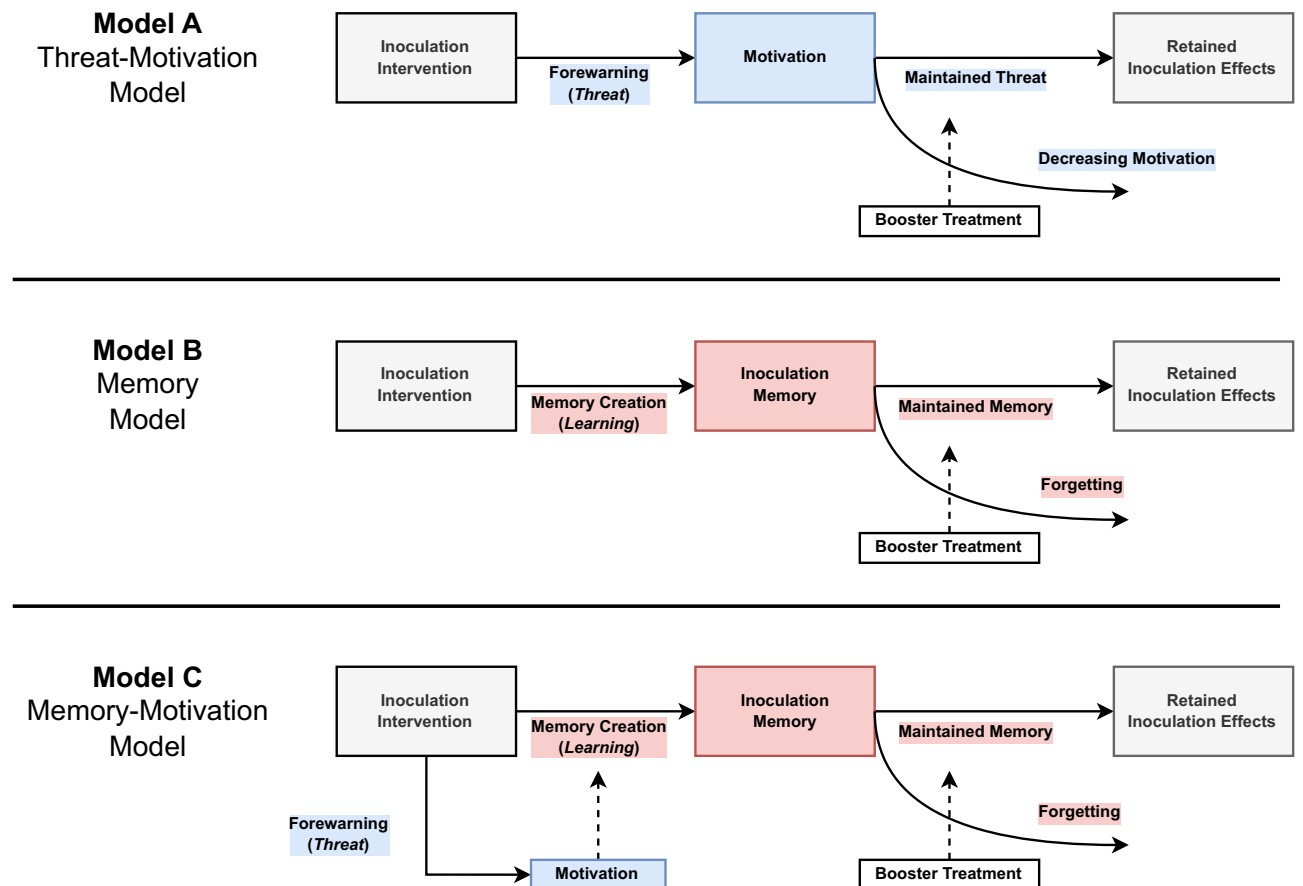

**Fig. 2 | Theoretical models explaining the long-term effectiveness of inoculation.** Three theoretical models to explain the long-term effectiveness of inoculation, a motivation-based model (Model **A**), a memory-based model (Model **B**), and a combined memory-motivation (Model **C**). Threat and motivation related variables are presented with a blue background. Memory-related variables are presented with a red background.

$d = 0.384$, 95% CI [0.209, 0.558], Bayesian posterior mean = 81.19, 95% CI [79.33, 83.04], BF10 = 877.955 (error = 0.783%). For the analyses at 29 days, we first found a significant omnibus test for the intervention variable, [H4] $F(2, 685) = 12.63$, $p < 0.001$, Bayesian posterior mean = 83.23, 95% CI [81.74, 84.72], BF10 = 2419.271 (error = 0.869%), and then found that the effect at 29 days was still significant with a smaller effect size, $t(685) = 2.96$, $p_{tukey} = 0.009$, $d = 0.281$, 95% CI [0.094, 0.468], Bayesian posterior mean = 5.34, 95% CI [1.08, 9.61], $BF_{10} = 2.061$ (error = 0.010%), reflecting that the effect had decayed but may still be present, although the Bayesian analysis shows only anecdotal support. As we expected the inoculation effect to no longer be significant after 29 days, the results are inconclusive for H4. See Fig. 3 (Panels A–B) for a visual plot of the inoculation effects over time.

Testing H5—whether the inoculation effect at T30 (29 days) was still significant for participants who took part in the booster intervention as compared to the control group—we found a significant, medium-sized effect, [H5] $t(685) = 5.02$, $p_{tukey} < 0.001$, $d = 0.475$, 95% CI [0.287, 0.662], Bayesian posterior mean = 9.07, 95% CI [5.09, 12.94], $BF_{10} = 2586.166$ (error <0.001%), in line with the hypothesis (see Fig. 3, Panels A–B). Although not preregistered, we also looked at the contrast between the booster group and the inoculation group, and found no significant effect, $t(685) = 2.13$, $p_{tukey} = 0.085$, $d = 0.194$, 95% CI [0.015, 0.373], Bayesian posterior mean = 3.67, 95% CI [−0.06, 7.41], $BF_{10} = 0.660$ (error = 0.029%).

We then tested the direct effects of the booster condition on the two mediators (memory, H6; motivation, H7) in the memory-motivation model at T30 (29 days). For this analysis we only looked at T30 as participants in the booster condition only received the posttest questions at T30 (at T0 and at T10 they received the

inoculation and booster interventions respectively without posttest measurement). We first found that the omnibus test for the intervention was significant for memory, [H6] $F(2, 686) = 95.28$, $p < 0.001$, Bayesian posterior mean = 7.09, 95% CI [6.94, 7.23], $BF_{10} = 8.113e + 33$ (error = 0.005%), in line with H6, but not for motivation, [H7] $F(2686) = 0.31$, $p = 0.731$, Bayesian posterior mean = 5.14, 95% CI [5.02, 5.26], $BF_{10} = 0.022$ (error = 0.028%), leading us to reject H7. The contrast between the double inoculation (booster) group and the single inoculation group, this time with objective memory as the dependent variable, showed a significant effect of the booster intervention, [H6] $t(686) = 8.15$, $p_{tukey} < 0.001$, $d = 0.741$, 95% CI [0.558, 0.924], Bayesian posterior mean = 1.35, 95% CI [0.99, 1.7], $BF_{10} = 2.816e + 10$ (error <0.001%). See Fig. 3 for a visualization of the effects of the intervention on memory (Panels C–D) and motivation (Panels E–F) as dependent variables over time, showing the memory boost at 29 days provided by a second inoculation after 8 days. Memory in this study is measured as the performance on a multiple-choice objective recall test of what was present in the original inoculation intervention (see Methods).

We then tested an approximation of the memory-motivation model using an SEM analysis with the *lavaan* package in $R^{34}$. In this model we included inoculation at T0 (yes/no) as a predictor variable, motivational threat and objective memory as mediator variables, and PSC as an outcome variable. We found that across the different time points, there were significant direct effects of memory and motivation on resistance to misinformation, and indirect effects of the inoculation intervention through memory and motivation. See Supplementary Fig. 1 for a schematic depiction of the T30 model, Supplementary Note 1 (incl. Supplementary Table 2) for an overview of the model estimates, Supplementary Note 2 (incl. Supplementary Table 3) for an

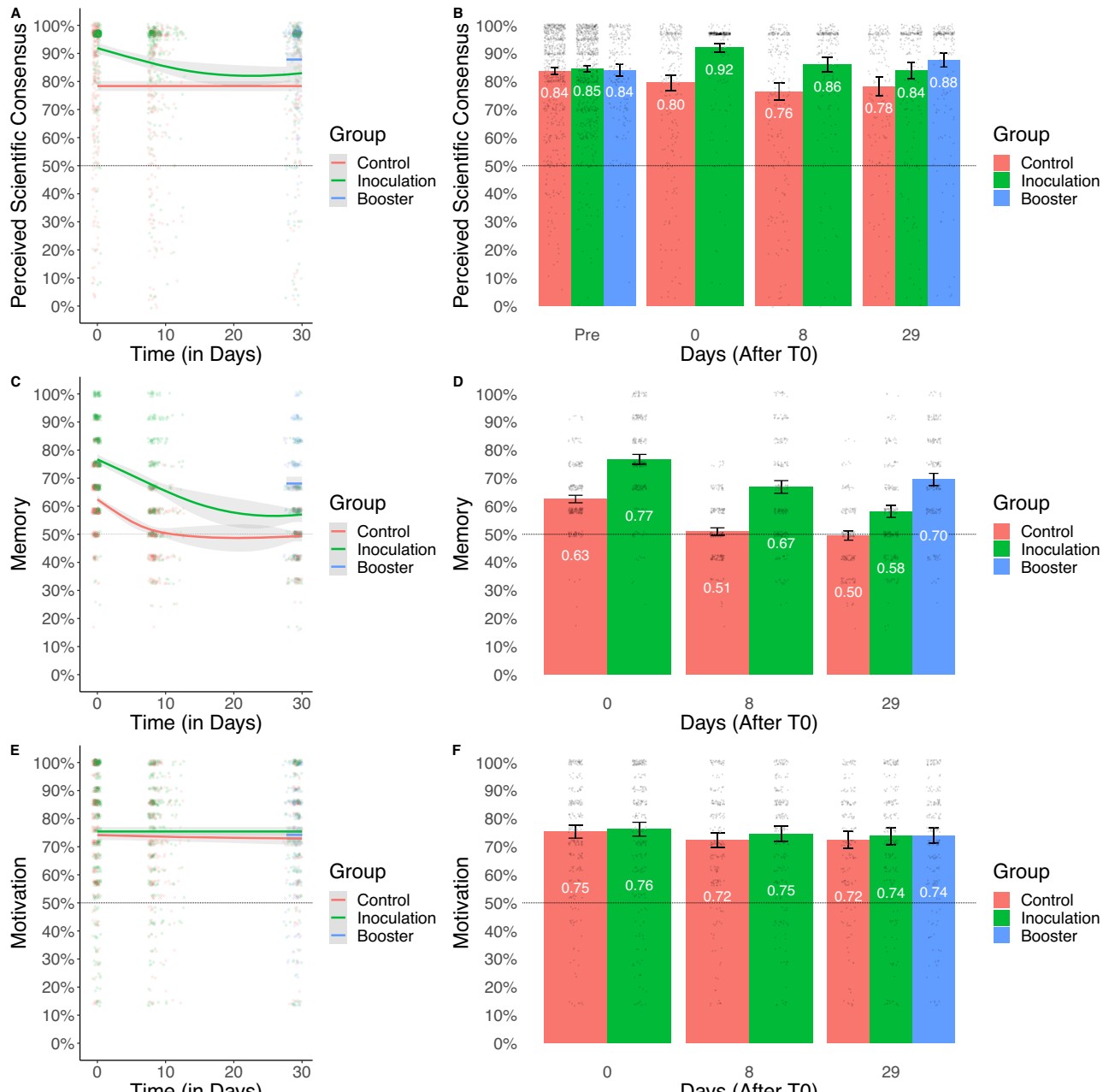

**Fig. 3 | Perceived scientific consensus, memory, and motivation over time for each group in study 1. A**, **C**, and **E** show the smoothed trends of perceived scientific consensus (representing the inoculation effect), memory (recall of the inoculation intervention), and motivation (to resist misinformation) over time, respectively, for three groups (Control in red, Inoculation in green, and Booster in blue). The error bands represent 95% confidence intervals around the mean. **B**, **D**, and **F** display the group means at specific time points (Pre, 0, 8, and 29 days after inoculation). The error bars represent 95% confidence intervals, and the center of each bar represents the mean. The sample size for the study is $N = 1825$.

exploratory analysis of the underlying mechanisms using dominance analysis, and Supplementary Fig. 2 for exploratory plots depicting the differential inoculation effects and decay curves in participant subgroups (incl. subgroups based on gender, misinformation susceptibility, political ideology, and age).

## Study 2: gamified inoculation

For Study 2 ($N = 674$) we implemented the gamified intervention design by Maertens et al.[28], and tested the Bad News game (BN; the game inoculates participants against six common misinformation techniques (e.g., polarization, conspiratorial reasoning)–rather than a single message–in a social media context using a broad range of topics

as examples) with the same approach and questions from Study 1 for memory and motivation, and a newly developed version of Bad News to serve as the booster game. We set out to shed further light on the validity of our memory theory of inoculation in the setting of gamified inoculation, and to investigate the potential of booster shots further. We sought to replicate the main effect at T0 (H1) and expected the long-term effectiveness to remain intact for at least 10 days (H2). Meanwhile, we expected the effect to no longer be significant after 30 days when no booster was received (H3), but still significant after 30 days if participants played a booster game 10 days after T0 (H4). We also expected the booster intervention to improve the objective memory of the intervention at T30 (H5), as well as increase the

motivation to defend oneself at T30 (H6). Finally, we aimed to test the importance of memory and threat at mediating the inoculation effect (H7). Memory in this study was measured by summing the scores on all objective multiple-choice memory test items for the original intervention ($M = 8.97$, $SD = 2.31$, $\alpha = 0.66$). Motivation was measured as the average rating of a series of subjective Likert-scale (1–7; $M = 5.19$, $SD = 1.33$, $\alpha = 0.81$) statements asking participants whether they were motivated to defend themselves against misinformation (see the printout on the OSF repository for a full overview of the threat and motivation measures).

We tested the main effect of the Bad News game on participants' reliability rating of misleading content using a one-way ANCOVA with pretest reliability ratings as a covariate, intervention as the independent variable, and misinformation reliability ratings as the dependent variable, at T0. We found inoculation to have a significant large effect on the outcome, [H1] $F(1, 316) = -43.37$, $p < 0.001$, $d = -0.779$, 95% CI [−1.020, −0.538], Bayesian posterior mean = 2.56, 95% CI [2.48, 2.63], $BF_{10} = 3.977e + 07$ (error = 1.152%), meaning that participants rated misinformation as less reliable after the inoculation intervention, and providing evidence in favor of H1 and replicating previous findings[28,35].

The same ANCOVA design as for H1 was used to test the decay hypotheses H2 and H3, this time at T10 ($Mdn = 9$ days after the intervention) and at T30 ($Mdn = 29$ days after the intervention). The inoculation effect was no longer significant 9 days after the intervention at the pre-registered alpha, [H2] $F(1, 239) = -3.54$, $p = 0.061$, $d = -0.244$, 95% CI [−0.500, 0.012], Bayesian posterior mean = 2.86, 95% CI [2.78, 2.94], $BF_{10} = 0.711$ (error = 1.189%), thereby providing no evidence for H2. Similarly, 29 days after the intervention, the omnibus ANCOVA test for the intervention was no longer significant [H3] $F(2, 325) = 2.64$, $p = 0.073$, thereby lacking evidence for an effect in the standard inoculation group (in line with our expectations for H3) nor an effect in the booster group (against our expectations for H4), but the Bayesian analysis [H3] posterior mean = 2.75, 95% CI [2.68, 2.83], $BF_{10} = 0.366$ (error = 1.755%), only showed anecdotal support for the null hypothesis and thus our conclusions for H3 and H4 are inconclusive. See Fig. 4 for an overview of the unreliability ratings (Panels A–B) over time.

As preregistered, we then continued to test whether the booster inoculation had a positive effect on memory of the T0 intervention (the total score on an objective test battery) and motivation at T30 (self-reported motivation to protect oneself against misinformation). For these analyses we used a T30 analysis of variance (ANOVA) similar to the one used for the previous hypothesis test but this time with memory and motivation as the dependent variable for H5 and H6, respectively, and without the pretest as a covariate. We first found that the intervention had a significant omnibus effect for memory, [H5] $F(2, 326) = 35.56$, $p < 0.001$, Bayesian posterior mean = 8.80, 95% CI [8.57, 9.03], $BF_{10} = 6.116e + 11$ (error = 0.017%), in line with H5, but not for motivation, [H6] $F(2, 326) = 0.06$, $p = 0.966$, Bayesian posterior mean = 5.15, 95% CI [5.01, 5.30], $BF_{10} = 0.034$ (error = 0.026%), leading us to reject H6. Looking at the specific group contrast for memory, we found a significant and large increase in memory for the booster intervention compared to the control group, $t(326) = 8.43$, $p_{tukey} < 0.001$, $d = 1.149$, 95% CI [0.867, 1.432], Bayesian posterior mean = 2.37, 95% CI [1.84, 2.89], $BF_{10} = 8.370e + 13$ (error <0.001%), in line with H5. Although not preregistered, we also looked at the difference in memory between the boosted inoculation group and the single inoculation group, also finding a significant difference, $t(326) = 3.99$, $p_{tukey} < 0.001$, $d = 0.538$, 95% CI [0.270, 0.806], Bayesian posterior mean = 1.08, 95% CI [0.54, 1.6], $BF_{10} = 348.383$ (error <0.001%). See Fig. 4 for a plot of the effect of the intervention on memory (Panels C–D) and motivation (Panels E–F) for each of the time points.

Our final hypothesis is a test of the memory-motivation model of inoculation, to investigate the interplay between memory and motivation in predicting inoculation effect outcome. To do this, as preregistered, we tested an SEM model that included inoculation as a predictor of the misinformation detection score, and memory and motivation as mediators. We found, in line with H7, that across time points, memory had a significant direct effect on misinformation reliability ratings, and the inoculation intervention had a significant indirect effect on misinformation reliability ratings through memory. The effects for motivation were not significant, except for the effect of motivation on memory. See Supplementary Table 4 for an overview of the preregistered hypotheses for Studies 3–5 and their evidence. See Supplementary Fig. 3 for a schematic presentation of the tested T0 approximation of the memory-motivation model, Supplementary Note 3 (incl. Supplementary Table 5) for a presentation and discussion of the relevant SEM model estimates, Supplementary Note 4 (incl. Supplementary Table 6) for an exploratory analysis of the underlying mechanisms using a dominance analysis, and Supplementary Fig. 4 for exploratory plots showing how the inoculation effects and decay curves differ in specific participant subgroups (e.g., differential effects depending on age).

## Studies 3–5: video-based inoculation

In Studies 3–5 we set out to explore the long-term effectiveness of video-based inoculation interventions (in which participants learn about using emotionally manipulative language using concrete examples and explanations that can be applied to a wide range of topics), as well as the mechanisms driving these effects. In our first experiment, we explored the effectiveness of a short inoculation video compared to a long inoculation video. In our second experiment, we tentatively explored the role of memory and motivational threat, as well as the longevity of the inoculation effect using multiple time points. In our third experiment, we attempted to replicate the findings from Study 4, test a memory-motivation model with a set-up comparable to Study 1 and Study 2, and explore the potential role of three types of booster interventions to determine which intervention most effectively boosts the inoculation effect. The booster videos were either a repetition of the original inoculation video (explaining how to spot manipulative emotional language and warning people of the threat of misinformation), a threat-focused video (that tried to boost threat and motivation by reminding people of the threat of misinformation, but did not reiterate how emotional language works), or a memory-boosting video (that reiterated how to spot manipulative emotional language but did not warn people of the threat). See Methods for an overview of how memory and motivation were measured. See Supplementary Table 7 for an overview of the preregistered hypotheses for Studies 3–5 and their evidence. For an overview of the analyses and test results for the hypotheses of Studies 3 and 4, see Supplementary Note 5 (incl. Supplementary Fig. 5) and Supplementary Note 6 (incl. Supplementary Figs. 6 and 7, and Supplementary Table 8) respectively.

We tested H3.1, which states that the short video improves manipulativeness discernment (whether people can distinguish manipulative social media posts from neutral ones), by exploring the main effect of the short inoculation video at T0, with the full sample size ($N = 5703$). We found that the effect was significant, [H3.1] $M_{diff} = 0.44$, $t(5701) = 8.76$, $p_{tukey} < 0.001$, $d = 0.295$, 95% CI [0.229, 0.361], Bayesian posterior mean = 0.44, 95% CI [0.34, 0.54], $BF_{10} = 1.199e + 15$ (error <0.001%).

After finding a significant omnibus test, [H3.2] $F(4, 2215) = 10.14$, $p < 0.001$, Bayesian posterior mean = 1.39, 95% CI [1.32, 1.45], $BF_{10} = 57779.737$ (error = 0.006%), we looked at the contrasts between the groups at T30. We found, contrary to our hypothesis H3.2, that the group which had not seen a repeated inoculation video or any of the two booster videos still showed a significant inoculation effect at T30 (29 days after T0), [H3.2] $M_{diff} = 0.34$, $t(2215) = 3.30$, $p_{tukey} = 0.009$, $d = 0.230$, 95% CI [0.093, 0.367], Bayesian posterior mean = 0.33, 95%

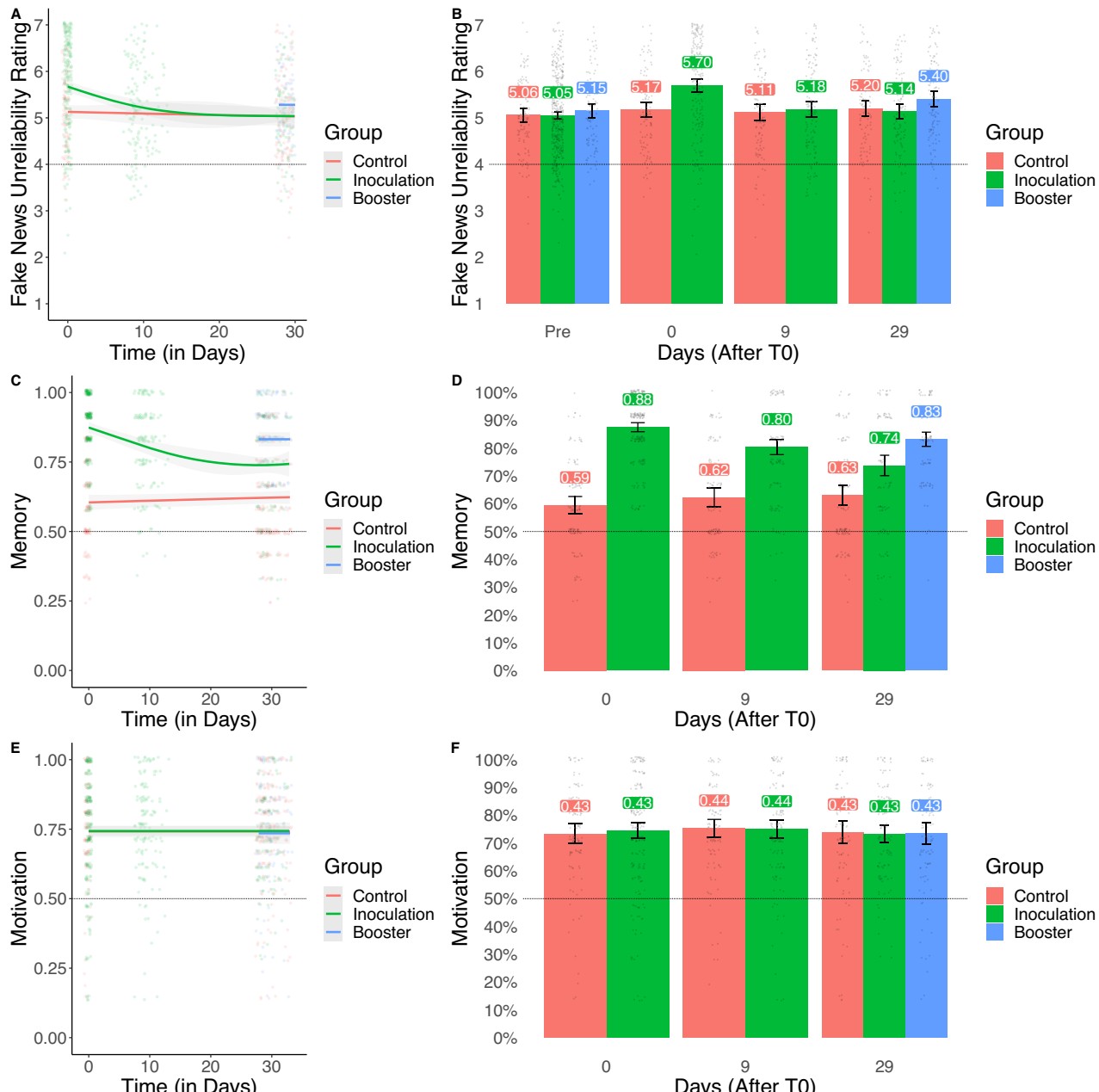

**Fig. 4 | Fake news unreliability ratings, memory, and motivation over time for each group in study 2. A, C,** and **E** show the smoothed trends of fake news unreliability ratings (representing the inoculation effect), memory (recall of the inoculation intervention), and motivation (to resist misinformation) over time, respectively, for three groups (Control in red, Inoculation in green, and Booster in blue). The error bands represent 95% confidence intervals around the mean. **B, D,** and **F** display the group means at specific time points (Pre, 0, 9, and 29 days after inoculation). The error bars represent 95% confidence intervals, and the center of each bar represents the mean. The sample size for the study is $N = 674$.

CI [0.14, 0.52], $BF_{10} = 26.187$ (error <0.001%). In line with H3.3, H3.4, and H3.5, we found that the inoculation effects remained significant for the groups that were boosted at T10 (9 days after T0), whether it was through a threat booster video, [H3.3] $M_{diff} = 0.38$, $t(2215) = 3.66$, $p_{tukey} = 0.002$, $d = 0.258$, 95% CI [0.120, 0.397], Bayesian posterior mean = 0.37, 95% CI [0.17, 0.57], $BF_{10} = 58.097$ (error <0.001%), a memory booster video, [H3.4] $M_{diff} = 0.64$, $t(2215) = 6.35$, $p_{tukey} < 0.001$, $d = 0.440$, 95% CI [0.303, 0.576], Bayesian posterior mean = 0.63, 95% CI [0.44, 0.82], $BF_{10} = 3.005e + 07$ (error <0.001%), or a repetition of the inoculation, [H3.5] $M_{diff} = 0.36$, $t(2215) = 3.65$, $p_{tukey} = 0.003$, $d = 0.250$, 95% CI [0.115, 0.384], Bayesian posterior mean = 0.36, 95% CI [0.16, 0.55], $BF_{10} = 69.409$ (error <0.001%). Descriptively, the memory booster video performs the best, with 100%

retention of the original effect size, while the re-inoculation condition and the threat booster conditions retain ~86% of the original effect size, and the control condition 78%. See Fig. 5 for a visual plot of the manipulativeness discernment (Panels A–B), memory (Panels C–D), and motivation (Panels E–F) in each condition over time in Study 5.

We then investigated the effect of booster sessions on the memory and motivation variables (H3.6–H3.9). The first three hypotheses were tested with motivation (model a, average rating on Likert-scale statements regarding motivation to protect oneself against misinformation; $M = 4.85$, $SD = 1.46$, $\alpha = 0.81$) or memory (model b, objective performance on a multiple choice test battery; $M = 7.86$, $SD = 2.52$, $\alpha = 0.57$) as the outcome variables. Model a, [H3.6a, H3.7a, H3.8a] $F(4, 2215) = 3.94$, $p = 0.003$, Bayesian posterior mean = 4.85, 95%

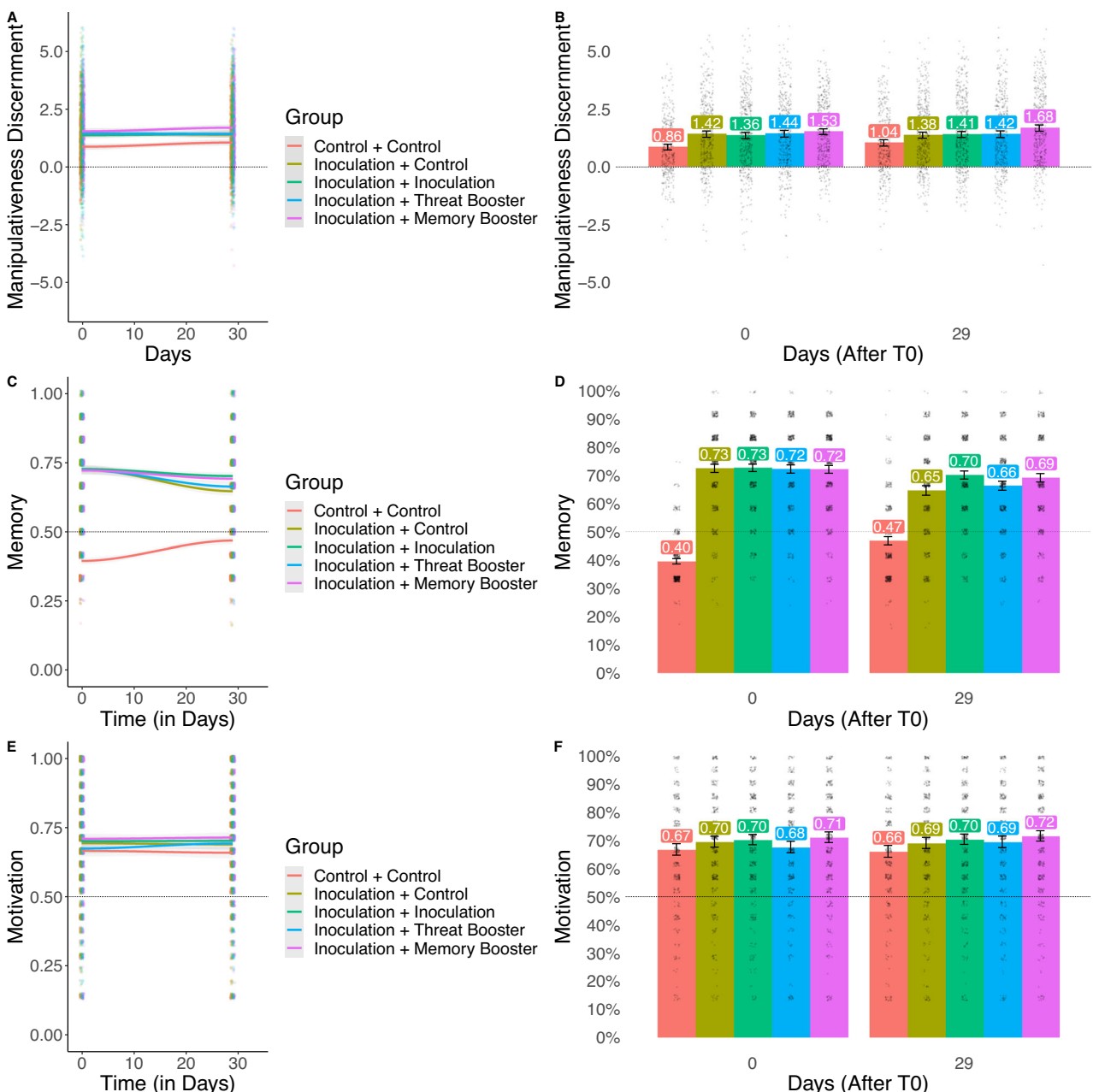

**Fig. 5 | Manipulativeness discernment, memory, and motivation over time for each group in study 5. A**, **C**, and **E** show the smoothed trends of manipulativeness discernment (representing the inoculation effect), memory (recall of the inoculation intervention), and motivation (to resist misinformation) over time, respectively, for five groups (Control + Control in red, Inoculation + Control in olive green, Inoculation + Inoculation in teal green, Inoculation + Threat Booster in blue, and Inoculation + Memory Booster in purple). The error bands represent 95% confidence intervals around the mean. **B**, **D**, and **F** display the group means at specific time points (0 and 29 days after inoculation), and follow the same color mapping as the panels on the left. The error bars represent 95% confidence intervals, and the center of each bar represents the mean. The sample size for the study is $N = 2220$.

CI [4.79, 4.91], $BF_{10} = 0.423$ (error = 0.042%), and model b, [H3.6b, H3.7b, H3.8b] $F(4, 2215) = 132.04$, $p < 0.001$, Bayesian posterior mean = 7.61, 95% CI [7.53, 7.70], $BF_{10} = 2.763e + 97$ (error = 0.012%), both showed a significant omnibus test, except for using the Bayesian analysis, which only provided evidence for model b. Looking at the preregistered contrasts, we found that a threat-focused booster video did not have a significant impact on motivation, [H3.6a] $M_{diff} = 0.03$, $t(2215) = 0.28$, $p_{tukey} = 0.999$, $d = 0.019$, 95% CI [−0.113, 0.151], Bayesian posterior mean = 0.03, 95% CI [−0.17, 0.23], $BF_{10} = 0.078$ (error = 0.252%), providing evidence against H3.6a, nor on memory, [H3.6b] $M_{diff} = 0.20$, $t(2215) = 1.51$, $p_{tukey} = 0.556$, $d = 0.102$, 95% CI [−0.030, 0.234], Bayesian posterior mean = 0.20, 95% CI [−0.08, 0.48],

$BF_{10} = 0.202$ (error = 0.100%), in line with H3.6b. Neither the memory-focused booster video, [H3.7a] $M_{diff} = 0.17$, $t(2215) = 1.81$, $p_{tukey} = 0.366$, $d = 0.120$, 95% CI [−0.010, 0.249], Bayesian posterior mean = 0.17, 95% CI [−0.02, 0.36], $BF_{10} = 0.377$ (error = 0.055%), nor the re-inoculation procedure, [H3.8a] $M_{diff} = 0.09$, $t(2215) = 0.97$, $p_{tukey} = 0.870$, $d = 0.063$, 95% CI [−0.065, 0.191], Bayesian posterior mean = 0.09, 95% CI [−0.10, 0.28], $BF_{10} = 0.115$ (error = 0.176%), had a significant effect on motivation, in line with H3.7a but inconclusive for H3.8a. Meanwhile both the memory-focused booster video, [H3.7b] $M_{diff} = 0.54$, $t(2215) = 4.14$, $p_{tukey} < 0.001$, $d = 0.273$, 95% CI [0.143, 0.403], Bayesian posterior mean = 0.54, 95% CI [0.27, 0.80], $BF_{10} = 152.974$ (error <0.001%), and [H3.8b] the re-inoculation procedure, $M_{diff} = 0.66$,

$t(2215) = 5.09$, $p_{tukey} < 0.001$, $d = 0.331$, 95% CI [0.203, 0.460], Bayesian posterior mean = 0.65, 95% CI [0.38, 0.91], $BF_{10} = 8403.542$ (error <0.001%), had a significant effect on memory, in line with both H3.7b and H3.8b.

Finally, to test H3.9, we implemented a SEM model similar to Studies 1 and 2, to test whether the effects of the intervention on the outcome variable are mediated by motivation and memory. We found evidence for full mediation of the inoculation effect through memory and motivation. See Fig. 6 for an overview of the memory-motivation model applied to Study 5 and Supplementary Note 7 (incl. Supplementary Table 9) for an overview and discussion of the model estimates, Supplementary Note 8 (incl. Supplementary Table 10) for a dominance analysis of the underlying mechanisms, Supplementary Note 9 (incl. Supplementary Fig. 8) for a word cloud analysis of the open memory questions in Study 5, Supplementary Fig. 9 for a plot of the inoculation effect across political leanings in the combined sample of Studies 3–5, Supplementary Fig. 10 for a plot of the inoculation effect across different memory groups at each time point across Studies 3–5, and Supplementary Fig. 11 (Sudy 3), Supplementary Fig. 12 (Study 4), and Supplementary Fig. 13 (Study 5), for exploratory plots showing how the inoculation effects and decay curves differ in specific participant subgroups (incl. subgroups based on gender, misinformation susceptibility, political ideology, and age).

## Discussion

Inspired by early research on the potential role of memory in inoculation interventions[29], we explored whether memory is an important mechanism in inoculation interventions, and whether in general interventions to counter misinformation could be extended by using booster interventions. Memory has long been studied in other areas of misinformation research, such as in the debunking literature[36]. Surveying the literature of preemptive counter-misinformation interventions made clear that knowledge about memory processes could help shed a new light on the underlying mechanisms of the longevity of interventions and how to extend their effects[37,38]. However, inoculation intervention designers have not yet tapped into the wealth of insights cognitive science and memory research can provide, such as predicting longevity based on a forgetting function, and making interventions more durable by making them more memorable or by using memory-boosting interventions. We integrated insights from the cognitive science of memory with the social science literature on countering misinformation, and proposed a memory-motivation model of inoculation (see Figs. 2 and 6).

Through a series of five studies, using three different interventions, we can now assess the validity and generalisability of a memory-based theory of the long-term effectiveness of inoculation. To measure the inoculation effects, participants reported their attitudes after a misinformation attack (Study 1) or were asked to rate the reliability (Study 2) or manipulativeness (Studies 3–5) of social media posts. We found that memory is one of the most dominant factors in intervention success and longevity. Given that this effect for memory was found when the inoculation and test stimuli were the same and fact-based (Study 1), when they were different and technique or logic-based (Studies 2–5), and when the inoculation was a video but the test stimuli involved text (Studies 3–5), there is an indication that the role of memory spans the scope of inoculation interventions and provides explanatory power across modalities. It also shows that people are able to flexibly apply remembered knowledge to new contexts and different stimuli—a concept known as cross-protection in inoculation theory[39]—which is essential in a world of fast-changing misinformation. Moreover, we found that booster interventions have the potential to further increase the longevity of intervention effects via memory strengthening. For text-based, gamified, and video-based interventions, we found that the effect shows a decay rate that is comparable to

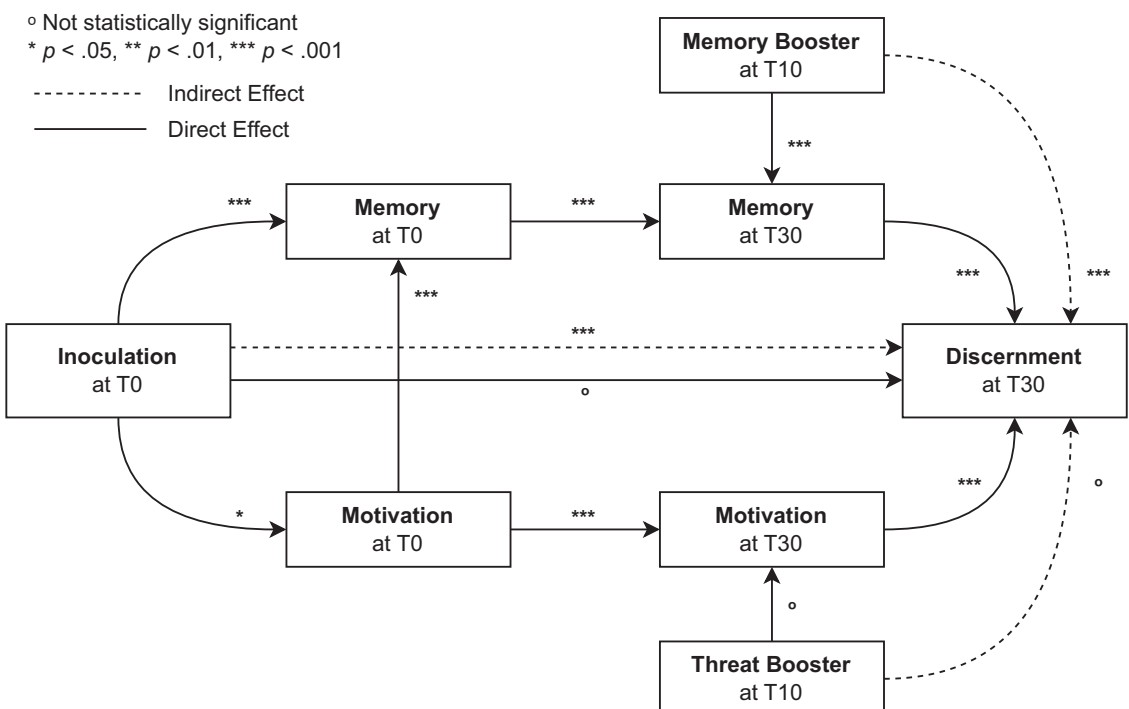

**Fig. 6 | The memory-motivation model of inoculation in study 5.** This figure represents the direct and indirect effects of inoculation on the discernment of manipulative items from neutral items (Discernment) at -30 days after the intervention (T30), through objective memory recall (Memory) and subjective motivation to defend oneself against misinformation (Motivation) immediately after the intervention (T0) and -30 days after the intervention (T30). It also depicts the effects of a memory-boosting intervention and a threat-boosting intervention administered -10 days after the intervention (T10). An equivalent model for text-based inoculation (Study 1) and gamified inoculation (Study 2) can be found in Supplementary Figs. 1 and 3 respectively. However, the model presented here is the most complete as it separates memory and threat boosters. $N = 2220$.

an exponential forgetting curve[40,41], and that the effect of specific text-based interventions can stay intact for about a month without a booster intervention, whereas the effects for the gamified and video-based interventions lost significance within the first two weeks without a booster intervention. This difference is likely due to the properties of specific issue-focused inoculations interventions, which are targeted to a limited amount of very specific content and may therefore be easier to remember and the attack stimuli easier to recognize—similar to the finding by Banas and Miller that interventions based on specific facts were more effective than logic-based interventions[42], and the meta-analytic finding that content-based inoculation yields larger effects than technique-based inoculation[16]. Meanwhile technique-focused inoculation interventions are broader and tap into multiple skills—as illustrated by the results of Studies 3–5 that showed the inoculation provided cross-protection to a wide range of stimuli and techniques—but therefore seem to be either less memorable, or make it more difficult to recognize the attack stimuli (i.e., marking it harder to activate memory of them). Basol et al.[43] for example found that text-based interventions decayed more quickly than gamified interventions when both are technique-focused. That said, the Bayesian analyses indicated that there was often only anecdotal evidence for the null hypothesis of the decayed inoculation effects at later time points, meaning more data would need to be collected for clarity on whether there is still a smaller inoculation effect left in the gamified and video-based interventions or if it reflects a true null effect.

Across the three inoculation formats, however, memory was consistently the most dominant outcome predictor, and booster shots consistently helped to restore intervention effects. Moreover, multiple forms of booster shots were shown to be effective: repeated interventions (see Study 1), new interventions targeting the same techniques (see Study 2), memory boosters (see Study 5), and quizzing participants in the form of posttesting (see results Study 3 vs Study 5). A threat-only booster intervention (see Study 5) did not seem to be effective, further strengthening the evidence for the dominant role of memory. It could be argued that a standalone affective threat booster had little chance of working as even if participants were going to be threatened, apprehensive, or motivated, it would benefit them to have *content* to process as well. In other words, the motivation and threat components of inoculation theory point towards threat being the motivational catalyst to *engage* in the resistance process[27], and therefore thinking through something, and as the memory-motivation model would propose, to learn something new and remember it. Nevertheless, as other scholars have noted, future research about the relevance of measuring affect at different stages of the inoculation remains an interesting avenue for future research[44].

A structural equation model analysis shows that across all five studies, using a model that integrates both memory and motivation provides a feasible and practical theory to map intervention effects, with motivation influencing the intervention memory and boosters strengthening it. In other words, the studies show consistently that the memory-motivation model proposed in Figs. 2 and 6 provides a valuable way to map and boost the longevity of counter-misinformation interventions (for a further discussion of these results, a more in-depth look at integrating memory research and inoculation theory, and a discussion on conceptual and methodological issues in longitudinal inoculation, see the Supplementary Discussion and Supplementary Fig. 14).

The finding of the dominant role of memory and an inoculation effect decay curve that is compatible with a memory forgetting curve could mean that multiple booster interventions may be needed to counteract misinformation in real world scenarios, but also that forgetting may flatten out when enough booster interventions are provided. In other words, in line with what would be expected from the memory literature, long-term retention could be achieved through repeated inoculation. This also fits within the biomedical inoculation metaphor, just like people may need multiple booster shots to foster immunity for COVID-19, which works in part by training memory B and T cells[45,46]. Future research should therefore explore repeated psychological booster shots. However, although the vaccine analogy is useful, we also recognize that the implication of comparing psychological inoculation to medical immunization could potentially be misinterpreted given the modest effects that also decay relatively quickly over time, so we would like to stress that psychological inoculation still differs from biological immunization in that it generally does not offer the same level of protection as a biomedical vaccine and wears off more quickly over time.

Some misinformation scholars have argued that in the days after an inoculation intervention, the inoculation effect might increase rather than decrease, as the inoculation effect might have to sink in refs. 47,48. However, our findings point in the opposite direction: decay is more likely to be exponential (i.e., more decay takes place closer to the intervention date). It is possible that the traditional theory—which posited the benefits of an initial period of delay, but has limited empirical evidence[30]—came into existence due to a lack of high-powered studies systematically looking at the decay curve and the mechanisms of decay. We propose an alternative theory to fill this gap: the memory-motivation model may complement the traditional inoculation model that was based on threat, motivation, and counterarguing, by adding a memory dimension to explain the long-term effectiveness of inoculation.

The series of studies presented in this work provide a response to three important theoretical, empirical, and methodological questions: (1) how long do the effects of counter-misinformation interventions based on inoculation last (i.e., what does the effect decay curve look like), (2) what are the mechanisms behind the (long-term) effectiveness, and (3) how can we boost the long-term effectiveness of these interventions taking into account the length of effects and the mechanisms of effect decay? By integrating insights from cognitive science with those from social psychology, we proposed a memory-motivation theory of resistance to persuasion by misinformation. In a series of five experiments using text-based, gamified, and video-based interventions, we unveiled the intervention effect decay function and established the importance of memory of the treatment in detecting misinformation, and provided evidence for the role of booster shots as a means to remedy forgetting. Additional evidence pointed towards motivation as a memory enhancer. We illuminated the underlying theoretical mechanisms of memory strengthening, finding that a regular booster treatment may be needed to enhance the inoculation effect by strengthening memory of the intervention. A comparison across three different media (text-based, gamified, and video-based), each utilizing different inoculation parameters, allowed us to determine the validity and generalisability of a memory-motivation theory of inoculation. The finding that short interventions can be as effective as longer interventions (see Study 3), indicates that it might be better for practitioners to focus resources on memory-boosting top-ups. This evidence for the dominant role of memory and the potential for booster shots in the longevity of inoculation effects contributes to our understanding of misinformation mechanisms and provides tools to those designing counter-misinformation policy and interventions, opening up opportunities to more effectively tackle misinformation.

## Methods
### Ethics and inclusion
The study protocols for Study 1 (ref. PRE.2021.086), Study 2 (ref. PRE.2021.087), and Studies 3–5 (ref. PRE.2021.012), were all reviewed and approved by the Cambridge Psychology Research Ethics Committee at the University of Cambridge prior to the start of data collection. Participants completed the study via an online Qualtrics survey in a self-directed manner, ensuring they were not influenced by the researchers during data collection. Informed consent was obtained

from the participants at the start of each study. The study goals were partly revealed at the informed consent stage of the study, but participants did not know all details nor the full purpose of the experimental conditions. The participants were in general blind to the exact experimental condition they were randomly allocated to. At the end of the study, the participants were debriefed and were given the opportunity to have their data removed, but no one asked for their data to be removed. All data has been anonymized prior to data analysis and publication. For each study we also recorded self-identified gender to ensure representativeness and balance in the data and included exploratory analyses for gender subgroups. All studies were conducted with participants based in the US, with the research team consisting of both local and international collaborators, including five UK-based authors and three US-based authors. Equally, the research team was diverse in gender and career, with five men and three women, and a combination of early career researchers, senior researchers, and practitioners.

## Study 1

**Intervention.** The first paradigm explores inoculation in the context of text-based climate change misinformation, for an overview see the Supplementary Methods (incl. Supplementary Fig. 15) and Maertens et al.[8]. This paradigm was chosen as (1) it is a well-established inoculation paradigm[7,20], (2) the topic is relevant for both theory (i.e., inoculation using a debated and polarized issue) and society, and (3) it can provide insights into the validity of the memory theory of inoculation when using a passive, specific, and therapeutic inoculation intervention.

**Design, sample, and procedure.** The study presents participants with a control task or an inoculation message (some participants see a repetition of this message after 10 days as a booster treatment), followed by—after a delay of 0 days (T0), 10 days (T10), or 30 days (T30)—a misinformation message about the scientific consensus on climate change (see the Supplementary Methods and Supplementary Fig. 15, for a detailed explanation of the intervention materials). The intervention is based on Maertens et al.[8], meaning that it includes the same misinformation, consensus, and inoculation messages, but for this study we added more and longer time periods, new motivation and memory measures, and a condition that includes a booster treatment. In addition, for this study, the consensus and inoculation messages were not but combined on a single page, and represent the inoculation group.

We recruited a high-powered sample (power = 0.95, $\alpha$ = 0.05, potential effect decay = 40%, attrition = 30%) of US participants aged 18 or older through Prolific ($N$ = 2657). The recruitment procedure was convenience sampling with equal balancing on gender, with a completion reward of ~£5.00 per hour (T0: £0.50, T10: £0.85, T10 booster: £0.60, T30: £0.85). As preregistered, participants were excluded when they (1) failed the manipulation check, (2) failed both attention checks, (3) participated in the survey multiple times, or (4) did not complete the entire survey. We also excluded participants who did not participate within a window of three days from the intended participation date (i.e., three days before or after). This led to a final sample size of $N$ = 1825, with an average of 260 participants per group, slightly below the intended $n$ = 328 due to a higher-than-expected attrition rate (T10$_{Attrition}$ = 24.34%, T30$_{Attrition}$ = 47.88%; with differences between the groups all being <10% from each other). Of the final sample, 49.21% identified as male (48.22% as female; 2.03% as non-binary; 0.33% as transgender, 0.22% as other), the average age was 35.79 ($SD$ = 13.07, $Mdn$ = 33), 58.69% had a higher education degree, 62.58% identified as left-wing (22.47% as centrist; 14.96% as right-wing), 48.99% identified most as Democrat (29.48% as Independent; 10.47% as Republican), 65.59% used social media multiple times a day (19.29% once a day, 7.29% weekly, 4.99% less often than weekly, 2.85% never), and 22.19% used Twitter multiple times a day (13.86% once a day, 12.06% weekly,

19.07% less often than weekly, 32.82% never). The participants were randomly allocated to one of three interventions: a word sorting task, the inoculation message, or the inoculation with a booster inoculation at 10 days. We also separated each time point by recruiting a separate sample for each condition, to avoid effects of repeated testing (i.e., each participant only ever received one post-test, at one time point depending on the group they were allocated to), leading to a total of 7 groups. The booster treatment employed in this study was an exact repetition of the original intervention. All participants received the misinformation message just before the posttest. When we refer to T0, if not otherwise specified, we refer to the posttest at T0. For a complete overview of the study design, see Supplementary Fig. 16.

**Materials and measures.** The main dependent variable for this study was the perceived scientific consensus on human-caused global warming, presented on a percentage slider scale ($M$ = 84.10, $SD$ = 16.77). Participants were asked to indicate, to the best of their knowledge, what percentage of climate scientists have concluded that human-caused climate change is happening (0% to 100%). The correct answer is 97%[49].

This study also introduced a new set of memory and motivation variables, as well as a range of measures related to inoculation effects and the memory-motivation model. Our main measure for memory was an objective, performance-based inoculation intervention content recall test, which we designed specifically for this study. The test included 12 questions, each with a single correct answer ($M$ = 7.54, $SD$ = 2.09, $\alpha$ = 0.57), consisting of 8 yes-or-no questions (for example, asking participants what they had learned from the messages presented in the first part of the survey, with response options such as false petitions, yes or no) and four multiple-choice questions (for example, asking participants about the topic of the messages in the first part of the survey, with response options such as: a—the scientific consensus on climate change, b—financial policy in the United States, c—the political side of bowling, d—vaccination intentions, e—none of these options is correct). These questions were combined into an index variable referred to as memory in this study (0–12; $M$ = 7.54, $SD$ = 2.09). For exploratory purposes, we also included a set of subjective memory measures created specifically for this study. These included self-reported remembrance, where participants rated how well they remembered the messages about climate change they saw earlier in the survey (on a Likert scale from 1 to 7, $M$ = 3.96, $SD$ = 1.86). In addition, four open-ended questions asked participants what they recalled from the first half of the survey, and three questions related to interference were combined into an interference index (for example, participants were asked whether they had heard conflicting arguments about climate change in the past two weeks, rated on a Likert scale from 1 to 7, with 1 being not at all true and 7 being very true; $M$ = 8.93, $SD$ = 4.66).

In addition to memory questions, we implemented a range of motivation measures. Our main measure for motivation was an adapted version of the motivational threat measure[50,51], which is considered the most predictive measure of threat-based motivation for inoculation-induced resistance to misinformation. We calculated this variable using a mean index of three Likert scale questions, such as participants rating the statement that thinking about climate change misinformation motivates them to resist misinformation on a scale from 1 to 7, with 1 being strongly disagree and 7 being strongly agree ($M$ = 5.20, $SD$ = 1.50, $\alpha$ = 0.90). As exploratory measures for the memory-motivation model, we also included measures for apprehensive threat (which differs from motivational threat by focusing on feelings of apprehension when exposed to a threat, rather than feeling motivated to respond adaptively, such as by counter-arguing), fear, issue involvement, issue accessibility, and issue talk[29,47,50–54], as described in Table 1.

The study was preregistered on 10th January 2022 at AsPredicted (https://aspredicted.org/b86tv.pdf). All statistical tests were

**Table 1 | Overview of exploratory measures of study 1**

| Construct | Type | M | SD |
|---|---|---|---|
| Apprehensive threat | Six Likert-scale questions (e.g., participants rated their agreement with statements about feeling threatened by climate change misinformation, on a scale of 1–7) | 3.92 | 1.73 |
| Fear | Mean index of three Likert scale questions (e.g., participants rated how fearful they felt about climate change misinformation, on a scale of 1–7) | 3.96 | 1.92 |
| Issue involvement | Index score from questions asking participants to choose the option that best describes how important climate change is to them, converted to a 1–7 scale | 6.27 | 1.76 |
| Issue accessibility | Single Likert scale item asking how often participants think about climate change, on a scale from 1–7 | 3.95 | 1.60 |
| Issue talk | Index of three questions, including Likert scale items on how often participants discussed climate change, and a choice list question about frequency of such discussions | 2.23 | 1.23 |

conducted using two-sided hypotheses, unless otherwise specified. We assessed normality and homogeneity of variances for all variables in all conditions for each study. Although not all tests indicated normality and homogeneity for all variables in all conditions, we chose to adhere to the preregistered analysis plan, as we prioritized transparency and consistency in our methodology. Nevertheless, we also added Bayesian analyses—which do not rely on strict parametric assumptions—as an alternative, and discuss transparently when the results do not align. A full list of the outcomes of the assumption tests, the original survey files, and a printout of the full survey, analysis scripts, and raw and clean datasets, are available on the OSF repository for this study at https://doi.org/10.17605/OSF.IO/9ZXJE[55].

**Deviations from preregistration.** We preregistered that we would exclude participants who did not participate in the follow-up within 5 days after the invitation. However, as the invitations were manual and grouped together, we invited participants 1–3 days earlier than the intended follow-up time. Therefore we have changed the exclusion window to 3 days before or after the intended follow-up time instead.

The analyses used for H3–H7 were slightly different from the preregistered analyses. The preregistration mentioned a repeated-measures ANCOVA but as we do not have fully balanced conditions and we have separate groups for each posttest time point (i.e., no repeated posttest measures), we cannot use a repeated-measures analysis. Instead, we use a separate ANCOVA for each time point to make up for this. Although not preregistered, we also report Bayesian analyses as an additional criterion, with analyses conducted using the BayesFactor package in $R$[56]. Throughout this work, we kept 0.707 as the Bayesian prior, representing an 80% chance that the effect size $d$ is between −2 and 2, as it allowed us to focus on both potential small and large effects, as well as to stick with the package's default for compatibility. We use the same models as for the frequentist analyses, applying the Bayesian equivalent of each model, in general consisting of omnibus AN(C)OVA tests for the intervention and $t$-tests for more specific comparisons. In addition, the posterior distributions are summarized using the posterior mean and 95% credible intervals. Sensitivity analyses to alternative priors were conducted, comparing priors of 0.707, 0.500, 0.350—based on the expected effect sizes as described in the preregistration—for all main text analyses, resulting in no major differences in the interpretation of the results. Spreadsheets with the full sensitivity analyses can be found on the OSF repository for each study (Study 1: https://doi.org/10.17605/OSF.IO/9ZXJE, Study 2: https://doi.org/10.17605/OSF.IO/HWMGE, Study 3–5: https://doi.org/10.17605/OSF.IO/ZRQ87)[55,57,58].

## Study 2
**Intervention.** The second paradigm used an interactive online inoculation game called Bad News, developed by Roozenbeek and van der Linden[22], in which people take the role of a fake news creator and spreader within a simulated Twitter-environment. To measure the effectiveness, participants rated the reliability of a set of social media

posts, and we looked at the reliability ratings of posts that made use of a misinformation technique. For an overview of the Bad News intervention and items, see the Supplementary Methods (incl. Supplementary Figs. 17 and 18) and Maertens et al.[28].

While the main intervention uses the same Bad News inoculation game as in Maertens et al.[28], we worked together with the media platform DROG to design a new, shortened, version of the Bad News intervention to serve as a booster treatment. In this 5-min version of Bad News, available at https://www.getbadnews.com/droggame_book/boostershot-bad-news/, participants are asked to put the skills they have learned in the original Bad News to use in a new scenario. They have to choose three disinformation techniques they want to revise and then have to use those disinformation techniques to go through an additional chapter, similar to the original Bad News.

We chose the Bad News paradigm for the second range of studies as it (1) describes an applied, implementable, and widespread intervention, and (2) to test the memory-motivation model in a broad-spectrum (i.e, it protects against a wide range of misinformation topics), interactive, inoculation intervention.

**Design, sample, and procedure.** We recruited 1350 US participants aged 18 or older through Prolific to participate in this study (based on a power = 0.95, $\alpha = 0.05$, accounting for up to 50% effect decay). The recruitment procedure was convenience sampling with equal balancing on gender, with a completion reward of ~£5.00 per hour (T0: £2.25, T10: £0.85, T10 booster: £0.60, T30: £0.85). Participants were randomly allocated to an inoculation group with a posttest at T0 only, an inoculation group with a posttest at T10 only (10 days later), an inoculation group with posttest at T30 only (30 days later), the booster group (with posttest at T30 only), or the control group (with posttest at T0, T10, and T30). Some participants in the inoculation group also received a booster treatment at T10. Participants at T0 also received a pretest to be used as a covariate during the study. When we refer to T0 in this study, when not otherwise specified, we refer to the posttest at T0. This design was chosen to avoid the boosting by repeated post-testing that we found in Maertens et al.[28] and enables a clean measure of the long-term effectiveness. We did not separate participants in the control group (i.e., every participant in the control group received all three posttest measurements) as previous studies had shown that the repeated testing effects in the control group were limited[28,59]. The time points were chosen to investigate the potential exponential decay between time points, and as we know from Maertens et al.[28] that the inoculation effect decays between T0 and 2 months later, and that the literature suggests that decay is likely to be found between 2 weeks[30,33] and 6 weeks[60]. The specific days between the recruitment were chosen to match the time points used in Study 1. See Supplementary Fig. 19 for an overview of the study design.

As preregistered, participants were excluded when they (1) failed the manipulation check, (2) failed both attention checks, (3) participated in the survey multiple times, or (4) did not complete the entire survey. We also excluded participants who did not participate in the

follow-up within 3 days from the intended participation date. This led to a final sample size of $N = 674$, with an average of 135 participants per group, considerably below the intended $n = 220$ due to a higher-than-expected attrition rate ($T10_{Attrition} = 33.03\%$, $T30_{Attrition} = 47.16\%$; with the differences between the groups all being <10% from each other), which may be in part responsible for why some of the hypothesized effects were descriptively going in the expected direction but not being significant, and why some of the Bayesian analyses showed only showed anecdotal evidence. Of the final sample, 54.30% identified as female (41.39% as male; 3.12% as non-binary; 1.04% as transgender, 0.15% as other), the average age was 33.18 ($SD = 12.25$, $Mdn = 30$), 53.12% had a higher education degree degree, 66.17% identified as left-wing (22.40% as centrist; 11.42% as right-wing), 68.55% used social media multiple times a day (17.66% once a day, 6.08% weekly, 4.75% less often than weekly, 2.97% never), and 24.63% used Twitter multiple times a day (15.88% once a day, 12.17% weekly, 21.66% less often than weekly, 25.67% never).

This study was also preregistered on 10th January 2022 on AsPredicted at https://aspredicted.org/gf5wx.pdf. The full results of all preregistered hypotheses and their evidence can be found in Supplementary Table 4. A full overview of all items and survey files, R analysis scripts, raw and clean datasets can be found at the OSF repository for this project at https://doi.org/10.17605/OSF.IO/HWMGE[57].

**Deviations from preregistration.** It was preregistered that we would exclude participants who did not participate within five days after the intended follow-up date. We chose to change the window to 3 days before or after that date as we sent out grouped invitations manually 1–3 days before the intended follow-up date.

The preregistration proposes a two-way repeated-measures ANCOVA analysis but the design of this study does not allow us to do this, as participants were separated in different groups for different time-points and the booster group did not receive a posttest before T30. We therefore use a one-way ANCOVA analysis for each time point separately and with pre-test as a covariate instead.

### Studies 3–5
**Intervention.** In Studies 3–5 we use a video-based inoculation paradigm. For a detailed description of this paradigm, see the Supplementary Methods (incl. Supplementary Figs. 20 and 21) and Roozenbeek, van der Linden, et al.[25]. We chose this final paradigm as it provides (1) a form of inoculation that is short and highly scalable, and (2) a test of the memory model for a broad-spectrum, passive (in contrast to the active Bad News intervention), inoculation intervention, enabling the further evaluation of the generalisability of the model.

**Procedure and measures.** After watching a video (which in the inoculation condition included an affective forewarning—the threat phase—and a technique training, both in function of teaching people how to recognize emotional-language-based misinformation; see https://inoculation.science/inoculation-videos/emotional-language/ for the full-length version of the inoculation video), participants completed a social media post rating task. This task involved rating a series of ten either manipulative (i.e., containing a manipulation technique) or neutral (i.e., not using any manipulation) social media posts that were based on actual news in the field[25,26]. The 10 headlines participants rated came from a pool of 20 items consisting of 10 pairs: for each news story we created a manipulative version and a non-manipulative version conveying the same message, and participants were randomly allocated a neutral or manipulative version of each pair, and all social cues (e.g., likes, names, sources) were redacted from the items. This also meant that the manipulative-to-neutral item ratio varied among participants. This setup allowed us to calculate a clean discernment index without the influence of topics, social cues, or item ratios. Specifically, participants were asked to indicate for each post (1) how manipulative they found the post (our main dependent variable for this study); (2) how confident they were in their ability to assess the post's manipulativeness; (3) how trustworthy they found the post; and (4) how likely they were to share the post with others in their network. This rating task was our main method of assessing the videos' efficacy in terms of improving participants' ability to identify manipulative content: if the inoculation videos are effective, treatment group participants should be significantly better than a control group at discerning manipulative from non-manipulative content, have significantly higher confidence in their ability to do so, find manipulative content less trustworthy than neutral content, and should display significantly less sharing intentions for manipulative content than for neutral content.

In addition, we investigated the underlying mechanisms of the inoculation effect in line with Study 1 and Study 2. We asked a set of questions to assess participants' sense of threat about emotional language on social media and related constructs[29,47,50–54], as well as our own battery of memory questions. See Table 2, the Introduction, and Study 1 (Methods), for a more detailed discussion of these measures. As an exploratory measure we also created a measure for concept mapping, in which participants had to write down as many concepts related to the theme and intervention as possible in open boxes (0–9; $M = 1.90$, $SD = 1.73$), inspired by the memory concept mapping method by Pfau et al.[29].

To explore further covariates we also measured misinformation susceptibility (as measured through the 8-item Misinformation Susceptibility Test or MIST-8; Maertens et al.[61]; 0–8; $M = 5.98$, $SD = 1.70$), conspiracy mentality (CMQ; Bruder et al.[62]; 1–7; $M = 4.58$, $SD = 1.30$), the level of trust in politicians, family members, journalists, and civil servants, party affiliation, political self-identification, and self-reported ideology in terms of social (1–7; $M = 3.96$, $SD = 1.74$) and economic (1–7; $M = 4.35$, $SD = 1.70$) issues. Finally, all participants responded to the same series of demographic questions: age, gender, education level, racial background, country of residence, news consumption behavior, whether English is their first language, and their favorite media outlet.

The Qualtrics files and the full PDF printout of the surveys can be found on the OSF repository for this study at https://doi.org/10.17605/OSF.IO/ZRQ87[58].

**Sample.** In Studies 3–5 participants were recruited and rewarded for their participation by Bilendi & respondi (an ISO-certified online panel provider with a proprietary incentive program where members earn and accumulate points for participating in surveys). All samples were representative inter-locked hard quota samples of the United States based on the age and gender composition data provided by the United States Census Bureau (2019). After recruitment and informed consent, participants took part in a Qualtrics survey and were randomly allocated to one specific condition, followed by a posttest, and in some cases a follow-up. All datasets, analysis scripts in R, Qualtrics surveys, preregistrations, and stimuli are available on the OSF repository at https://doi.org/10.17605/OSF.IO/ZRQ87[58].

**Study 3 specifics.** The goals of this study were as follows: (1) to replicate the effect of the emotional language inoculation video from Roozenbeek, van der Linden, et al.[25], (2) to identify differential effect sizes depending on video length (the full-length 1:48 min video and its shorter version of 0:30 min), (3) to determine the decay percentage after a two-week period, (4) to explore the role of memory and threat in inoculation effects, and 5) to explore if the inoculation effect is moderated by covariates such as conspiratorial thinking, misinformation susceptibility, and political polarization. To answer these questions, we conducted a preregistered longitudinal randomized controlled trial with power = 0.95, $\alpha = 0.05$, and attrition of 30% for an effect size of $d = 0.490^{\text{based on }25}$. The recruited sample size was $N = 2895$, with a reduction to $N = 2219$ when counting complete responses only

**Table 2 | Overview of exploratory measures of study 3**

| Construct | Type | M | SD |
|---|---|---|---|
| Objective memory | Index (0–12) of four multiple-choice questions (e.g., an example item asks what example was given in the video for using emotional language in news headlines, with options such as changing a headline from serious accident to horrific accident, using a radio broadcast to trigger emotions, or employing emojis to trigger emotions) and eight yes-or-no questions (e.g., participants are asked which of the following they learned about in the video, with answer options like the role of fear and outrage, yes or no) | 7.31 | 2.43 |
| Self-reported remembrance | Single-item Likert scale (1–7); example item: participants are asked how well they remember the video shown at the beginning of the survey, with ratings from 1 (I remember nothing) to 7 (I remember everything) | 3.61 | 1.90 |
| Self-reported interference | Index (1–7) of three Likert items; example item: participants are asked how often, in the past two weeks, they have seen videos about emotional language, with ratings from 1 (Not at all true) to 7 (Very true) | 2.76 | 1.61 |
| Motivational threat | Index (1–7) of three Likert items; example item: participants are asked how much the idea of emotional language on social media motivates them to resist misinformation, with responses ranging from 1 (Strongly disagree) to 7 (Strongly agree) | 4.88 | 1.45 |
| Apprehensive threat | Index (1–7) of seven Likert items; example item: participants are asked how threatened they feel by emotionally manipulative language on social media, with responses ranging from 1 (Strongly disagree) to 7 (Strongly agree) | 3.32 | 1.67 |
| Fear | Mean index of three Likert items; example item: participants are asked how fearful they feel about emotionally manipulative language on social media, with responses ranging from 1 (None of this feeling) to 7 (A great deal of this feeling) | 3.00 | 1.77 |
| Issue involvement | Index (1–7) of six choose-one-option-from-this-pair questions, asking which option of each pair best describes how much deception by emotionally manipulative language on social media means to them, with ratings ranging from insignificant to significant | 4.81 | 2.49 |
| Issue accessibility | Index (1–7) of two Likert items; example item: participants are asked how often, compared to other issues, they think about the issue of manipulative news (e.g., using emotional language), with responses from 1 (Never) to 7 (Very often) | 3.70 | 1.60 |
| Issue talk | Index (1–7) of three questions, including two Likert scale items on how often participants talked about the issue of emotional language on social media in the past two weeks (1 = Never, 7 = Very often) and two choice-list questions on how many times participants discussed the issue of manipulative news in the past two weeks (e.g., 0, 1, 2, 3, 4, 5, More than 5) | 2.37 | 1.38 |

and after—as preregistered—removing participants who failed both the manipulation and the attention check, participated multiple times, or entered the same response to each of the items of the dependent variable. The attrition rate for those allocated to the follow-up condition was 36% (slightly higher than preregistered), but the largest difference in attrition between conditions was less than 5%. In our final sample, 50.70% identified as female (48.26% as male; 0.86% as non-binary; 0.05% as other; 0.14% preferred not to answer), the average age was 46.00 ($SD = 16.41$, $Mdn = 46$), 66.70% had a higher education degree (1.85% did not finish high school), 31.73% identified as left-wing (32.85% as centrist; 35.42% as right-wing), 37.72% identifies most as Democrat (29.61% as Independent; 30.28% as Republican), 39.84% checked the news multiple times a day (34.84% once a day; 14.65% weekly; 8.43% less often than weekly; 2.25% never), 54.08% uses social media multiple times a day (23.43% once a day; 9.55% weekly; 5.36% less often than weekly; 7.57% never), 29.11% used YouTube multiple times a day (22.89% once a day; 26.32% weekly; 16.09% less often than weekly; 5.59% never), and 6.17% uses YouTube for news consumption multiple times a day (12.89% once a day; 16.54% weekly; 23.98% less often than weekly; 40.42% never). In Study 3, the rating task was administered at two different time points: T0 (immediately after watching the video) and T10 (two weeks after watching the video). Participants were randomly assigned to one of six conditions (see Supplementary Fig. 22 for an overview): the short inoculation condition (with posttest at T0 or at T10), the long inoculation condition (with posttest at T0 or T10), or the control condition (with posttest at T0 or at T10). The rationale for this design, and specifically for the splitting in different sample groups per posttest time point, is to eliminate repeated testing effects, which could lead to unwanted effect-boosting confounds in the measurement of decay[28]. Note that T0 represents the day of the intervention, and as there was no pretest, we refer to T0 as the posttest at T0 (unless otherwise specified). This study was preregistered on 18th June 2021 on the AsPredicted platform at https://aspredicted.org/fs2ph.pdf, and all analysis scripts in *R*, items, and Qualtrics survey files can be found on the OSF repository at https://doi.org/10.17605/OSF.IO/ZRQ87[58].

**Study 4 specifics.** The basics of the video-based inoculation paradigm, including the dependent variables, are the same as in Study 3. New in Study 4 is that we include only the short videos (0 min 30 s), have a larger sample size, and include multiple time points (4, 10, and 30 days). In total, we recruited $N = 5191$ participants to T0, with random allocation to each condition (see Supplementary Fig. 23 for an overview). After—in line with the preregistration protocol—removing participants that failed both the manipulation check and attention check, participated in the survey more than once, entered the same response to all items of the dependent variable, or did not complete the entire survey, a total of $N = 4821$ participants remained. Of our final sample, 51.73% identified as female (47.65% as male; 0.44% as non-binary; 0.10% as other; 0.08% preferred not to answer), the average age was 45.79 ($SD = 16.46$, $Mdn = 45$), 65.63% has a higher education degree (1.35% did not finish high school), 30.47% identifies as left-wing (35.51% as centrist; 34.02% as right-wing), 35.86% identifies most as Democrat (32.03% as Independent; 29.25% as Republican), 36.57% checks the news multiple times a day (35.72% once a day; 14.87% weekly; 9.83% less often than weekly; 3.01% never), 51.05% uses social media multiple times a day (25.16% once a day; 10.81% weekly; 6.16% less often than weekly; 6.82% never), 27.11% uses YouTube multiple times a day (22.59% once a day; 27.84% weekly; 16.74% less often than weekly; 5.73% never), and 5.83% uses YouTube for news consumption multiple times a day (11.28% once a day; 15.25% weekly; 22.13% less often than weekly; 45.51% never). Participant attrition levels were lower than the predicted percentages: 24.12% for T10, 27.68% T30, and 39.67% for T4 (with at each time point <5% difference between groups). The preregistration of this study was submitted on 8th September 2021 to AsPredicted but contained a copy paste error (https://aspredicted.org/zr8y3.pdf); the preregistration without the error was re-uploaded on 4th July 2022 and can be found on https://osf.io/av7zc. All analysis scripts in *R*, items, and Qualtrics survey files can be found on the OSF repository at https://doi.org/10.17605/OSF.IO/ZRQ87[58].

**Study 5 specifics.** In Study 5 we built further on the design of Study 4, as well as Study 1 and Study 2, by combining multiple videos to test

booster effects over time. In this final study we aimed to test and disentangle the two effects that drive inoculation effects: the threat component, and the preemptive refutation[17,27]. All participants were exposed to two different videos, a first video at T0, and a second video at T10 (*Mdn* = 9 days later). The first video was either the control video or the short inoculation used in Study 4. The second video was the same control or inoculation video repeated, a threat booster video focused on increasing levels of threat and motivation, or a memory booster video focused on reminding people of what they learned in the original intervention. We designed the threat booster video in such a way that it employed emotional music and warned people about manipulative online content, but it did not explain the methods that are used to mislead people nor use any of the content from the original video (i.e., only threat, no refutational preemption). The memory booster, by contrast, omitted the emotional music and affective forewarnings, but it did repeat the explanation of the techniques that can be used to mislead people using emotional language with similar content to the original video. Finally, all participants took the manipulativeness discernment test at T0 and at T30 (*Mdn* = 29 days later). This allowed us to disentangle and link effects at immediate posttest and at later posttest to enable testing the memory-motivation model. All participants were randomly allocated to the different video combinations (see Supplementary Fig. 24 for an overview).

In total, we recorded 6164 survey responses at T0. As preregistered, we excluded incomplete and low-quality responses, leading to a T0 sample size of 5703. Finally, we removed participants that did not participate in all three parts of the survey or did not participate in the follow-up sessions within 3 days before or after the intended time (T10: 10 days after, T30: 30 days after). This led to a final sample size of 2220, with an average of 444 participants per group. This is slightly below but close to the intended 548 participants per group (participant attrition from T0 to T30 was 60%, slightly above the estimated 55%; with all between-group differences in attrition being less than 10%). In our final sample, 55.14% identified as female (44.50% as male; 0.23% as non-binary; 0.09% as other; 0.05% preferred not to answer), the average age was 53.29 (*SD* = 14.48, *Mdn* = 55), 67.48% has a higher education degree (1.40% did not complete high school), 29.19% identifies as left-wing (34.23% as centrist; 36.58% as right-wing), 36.13% identifies most as Democrat (28.87% as Independent; 32.07% as Republican), 40.90% checks the news multiple times a day (36.85% once a day; 12.52% weekly; 7.70% less often than weekly; 2.03% never), 45.68% uses social media multiple times a day (25.09% once a day; 10.90% weekly; 7.21% less often than weekly; 11.13% never), 21.89% uses YouTube multiple times a day (19.77% once a day; 29.59% weekly; 21.08% less often than weekly; 7.66% never), and 5.72% uses YouTube for news consumption multiple times a day (9.86% once a day; 11.49% weekly; 21.62% less often than weekly; 51.31% never). This study was preregistered on 17th November 2021 on the AsPredicted platform at https://aspredicted.org/tf5g7.pdf, and all analysis scripts in *R*, items, and Qualtrics survey files can be found on the OSF repository at https://doi.org/10.17605/OSF.IO/ZRQ87[58].

**Deviations from preregistration.** For Study 3, there are no deviations to report. For Study 4, we preregistered a repeated-measures ANOVA analysis but this approach did not allow us to test the planned contrasts, as the immediate posttest would be used as a covariate. Therefore, we used a standard ANOVA analysis instead. Similarly, for Study 5, we were not able to execute the planned fully crossed repeated-measures ANOVA analysis as the design is not fully balanced. We therefore have opted to use a standard ANOVA analysis in Study 5 as well.

**Reporting summary**
Further information on research design is available in the Nature Portfolio Reporting Summary linked to this article.

## Data availability

The supplements, raw datasets, clean datasets, survey printouts, pre-registrations, and materials related to this work have been deposited in the Open Science Framework (OSF) repositories for this project, where they are publicly accessible at https://doi.org/10.17605/OSF.IO/7WQZT (overview), https://doi.org/10.17605/OSF.IO/9ZXJE (Study 1), https://doi.org/10.17605/OSF.IO/HWMGE (Study 2), and https://doi.org/10.17605/OSF.IO/ZRQ87 (Studies 3–5)[55,57,58,63].

## Code availability

The Qualtrics survey configuration files, R cleaning scripts, and R analysis scripts for this work have been deposited in the Open Science Framework (OSF) repositories for this project, where they are publicly accessible at https://doi.org/10.17605/OSF.IO/7WQZT (overview), https://doi.org/10.17605/OSF.IO/9ZXJE (Study 1), https://doi.org/10.17605/OSF.IO/HWMGE (Study 2), and https://doi.org/10.17605/OSF.IO/ZRQ87 (Studies 3–5)[55,57,58,63]. All experiments were set up and run online using Qualtrics (v2021, v2022). All data analyses were done in R (4.0, 4.1, 4.2, 4.3, 4.4), with the following additional packages: afex (1.4.1), BayesFactor (0.9.12.4.7), car (3.1.2), cowplot (1.1.3), domir (1.2.0), dplyr (1.1.4), effectsize (0.8.9), emmeans (1.10.4), ggh4x (0.2.8), ggplot2 (3.5.1), jamm (1.2.4), jmv (2.5.6), jmvcore (2.6.3), lavaan (0.6.18), NLP (0.3.0), openxlsx (4.2.7), plyr (1.8.9), psych (2.4.6.26), purrr (1.0.2), RColorBrewer (1.1.3), reshape2 (1.4.4), Rmisc (1.5.1), scatr (1.0.1), semPlot (1.1.6), semTools (0.5.6), SnowballC (0.7.1), stats (4.4.0), tibble (3.2.1), tidyr (1.3.1), tm (0.7.14), wordcloud (2.6), writexl (1.5.0).

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

## Acknowledgements

This work was financially supported by the University of Cambridge Department of Psychology, the IRIS Infodemic Coalition (UK Government; #SCH-00001-3391), and Google Jigsaw (Google LLC). R.M. was financially supported by the United Kingdom Economic and Social Research Council (ESRC; #ES/P000738/1), Cambridge Trust (CT), and Google Jigsaw (Google LLC). S.L. acknowledges financial support from the European Research Council (ERC Advanced Grant 101020961 PRODEMINFO), the Humboldt Foundation through a research award, the European Commission (Horizon 2020 Grant 101094752 SoMe4Dem), and the UKRI (via EU Horizon replacement funding grant number 10049415). S.v.d.L. received funding from the UK government Infodemic Coalition (#SCH-00001-3391) and Google Jigsaw (Google LLC). We also would like to thank Cecilie Steenbuch Traberg for her help with creating the stimuli (social media posts). We also want to thank Luke Newbold, Sean Sears, and Studio You in London for creating the videos. Finally, we would like to thank DROG, TILT, and Gusmanson Design for helping create the Bad News game.

## Author contributions

R.M., J.R., J.S.S., S.L., V.M., B.G., and S.v.d.L. all contributed to designing the project, research materials, and experimental designs. R.M. led the work on initial conceptualisation, project management, data collection, and data analysis, and wrote the first version of the manuscript. R.M., J.R., J.S.S., S.L., V.M., B.G., R.X., and S.v.d.L. all contributed to the review and revision process, improving the writing, data interpretation, and literature analysis. S.v.d.L. had an additional responsibility for supervising the project and ensuring the quality of the work.

## Competing interests

The authors declare no competing interests.
