## [Transparent Peer Review file · Nature Communications]

Psychological Booster Shots Targeting Memory Increase Long-Term Resistance Against Misinformation

Corresponding Author: Dr Rakoen Maertens

Version 0:

Reviewer comments:

Reviewer #1

(Remarks to the Author)

The present manuscript describes five longitudinal experiments investigating the “long-term” effectiveness of psychological “inoculation”. The research addresses a long-needed aspect in the literature on inoculating against misinformation: A systematic assessment of its long-term effects. I applaud the authors for transparent reporting of results, including deviations from preregistration. However, I also have a couple of concerns, three major and several smaller ones, detailed below.

Major issues:

- The biggest point is this: Transparency and fairness in using the vaccination metaphor after the present results. Given the small effects of inoculation over time (e.g., day 30 in Figure 3), I wonder whether it is fair and accurate to still use the “inoculation” metaphor so broadly. One decisive feature of actual vaccinations is precisely their longevity, see the example of childhood vaccinations. This is clearly not the case here, or at least not under all the conditions where the metaphor is applied to. Given the partially small effect sizes after just one month, my honest opinion is this: I do not think it is fair to exploit that vaccination metaphor any longer. For the sake of transparency and honesty in reporting, especially to the general public, what the authors refer to as “inoculation” should simply be described as “providing information” or “gaming”, even if this means sounding less flashy. This is not only the case in order to fairly describe the long-term effects (or lack thereof) of these “inoculations” but also to make sure that the public does not come to believe that the effectiveness of actual vaccinations suffer from such dramatic decay rates, rather being (demonstrably!) effective for years or even decades.
- Similarly, the title refers to “long-term” resistance. This is simply not fair, to put it mildly. No vaccination would be considered to be effective in the “long run” if it demonstrably works for 30 days. One month is surely longer than what is revealed in typical cross-sectional analyses, but it is surely not what counts as “long-term” with respect to vaccinations per se. Please tone down, even if only in the sake of not committing to the same errors as those you mean to inoculate against—after all, click-bait headlines only defeat your very purpose.
- It would be preferable to report Bayesian statistics as opposed to frequentist results. This is for several reasons, among them providing more robust estimates and providing evidence for the Null.

Minor points:

- under: The Present Research: It is unclear what “these mechanisms” refers to
- in how far constitutes the spreading of own misinformation in Bad News a “severely weakened form”?

Reviewer #2

(Remarks to the Author)

This paper describes five experiments assessing long-term effectiveness of “inoculation” interventions for misinformation, and evaluating the role of memory for the intervention and motivational threat in the longevity of the effect. I appreciate the authors’ efforts to bring several related studies together to tell an overarching story about the effectiveness of inoculation. The five included studies are however very different from one another in rationale and methodology, and many important nuances are lost in the effort to find commonalities. Overall, I believe this paper can make a valuable contribution to the literature, but some clarifications are needed. I have listed a few additional points below.

1. I appreciate that this is not entirely within the authors' control, but I found the structure of the paper very hard to follow. Methodological details are split across the Results section, the Methods section, the (very long) supplement and the As Predicted preregistrations. For other details, we are directed to previous papers (e.g. Maertens et al., (2020)). In order to understand exactly what was happening at any given point, I had to continually flick back and forth between all of these sources, and I am not confident that I have absorbed all the details correctly. Apologies if this leads to errors in any of my review points. To remediate this, I suggest including more methodological detail in the main text if possible; for example, information about the nature of the inoculation videos in studies 3-5 and the topic of misinformation in studies 2-5 is buried deep in the supplement, and I don't believe the "false petitions" element is explained at all.
2. It is often difficult to establish exactly which effects are being discussed. For example, on page 9: "We first found that the inoculation effect was still significant at 8 days". It is not clear what is meant by "inoculation effect" here. Is this simply a comparison between T1 and T10 within the inoculation group, or is it the difference between the inoculation and control group? These issues are present throughout the paper, but in most cases could be resolved by reporting the full ANOVA/ANCOVA results, rather than selected terms. These data might be best presented in table format.
3. I may be missing some important context here (I recognise that the authors are constrained by a tight word limit) but as written it seems that the control group is not being used as a comparator in the assessment of inoculation effects. Please clarify if this is the case, and if so, explain why. It seems like the interaction between group and time would be the most important term here as it would allow the authors to determine whether performance in the inoculation group(s) exceeds that in the control group at each time point.
4. Study 1: The authors used two paired t-tests to separately examine the misinformation effect for the control and inoculation groups. They report a significant effect for the control group, but not for the inoculated group. I was surprised that the authors did not use a 2x2 ANOVA here as this approach does not allow inferences regarding the relative performance of the two groups (i.e. the fact that there was a misinformation effect in one group but not the other does not imply that the two groups differ significantly from each other). This comparison is embedded within the 3-way ANOVA, but this analysis is described so briefly that I had trouble following the interpretation.
5. Please include pre-test ratings of scientific consensus in Figure 3 for comparison purposes.
6. The motivational threat measure was not defined in the preregistration, and a number of different measures appear to have been selected here. Please provide further details about how the three items included in this measure were selected. The difference between "motivation threat" and "apprehensive threat" is not explained.
7. Please provide details of the validity and reliability of the memory and threat scales. Were pilot data obtained for these measures?
8. Study 2: Please remove references to trends in non-significant findings providing 'mixed evidence' for hypotheses or being "on the border of significance". This also applies to Table S24 in the supplement, where clearly non-significant findings are listed as 'mixed'. The analyses were well-powered but could not provide support for the hypotheses at the preregistered alpha level, so these claims are misleading.
9. Studies 3-5: Please clarify the primary outcome variables in the main text. This section refers to "discernment", which implies the ability to discriminate between true and false news, but the method indicates that this is in fact "manipulativeness discernment", which appears to measure whether or not participants can identify particular elements in a video. It is important not to conflate this with misinformation detection, since manipulative elements can be present in truthful information.

Reviewer #3

(Remarks to the Author)

I have completed by review of NCOMMS-23-14656-T, entitled "Psychological Booster Shots Targeting Memory Increase Long-Term Resistance Against Misinformation." The manuscript reports five pre registered longitudinal experiments that examine the effectiveness of various psychological inoculation interventions with nearly 12,000 participants in total. Overall, I am quite impressed with the totality of research in this manuscript. As with any study, I find some positive aspects especially noteworthy, and I have some suggestions for improvement.

Positives:

1. The topic of a misinformation is one of crucial importance, and I applaud the authors for researching this issue. As noted in the manuscript, discerning misinformation isn't just essential for adaptive decision making at individual and societal levels, the proliferation of misinformation has directly contributed to violence and death.
2. I think the authors make a strong case for messaging that is preemptive in nature, and I think studying inoculation theory in the context of addressing misinformation is very sensible.
3. The most impressive element of the manuscript for me is the testing of several different competing and complementary theoretical ideas across a number of studies. Studying the mechanisms of inoculation is a longstanding tradition in the literature, and looking at the roles of motivation and memory, especially as to how they may both mediate the inoculation process to affect misinformation discernment in the proposed memory-motivation model, is a valuable step forward for the

inoculation literature.

4. In addition to the mechanisms, the focus on temporal dynamics of inoculation interventions also addressed other theoretical issues, like the related issues of message decay and booster shots. The role of message decay is of theoretical importance but also practical importance is misinformation interventions are actually to have a pronounced effect on the problem. I liked how the authors tied connected boosters and decay back to the research on memory.

5. In addition to the clever theorizing, the studies were carefully conducted and analyzed. And the Ns were impressive.

Areas for improvement:

1. I believe Pfau et al. (2005) were not arguing against threat as much as arguing that memory and mental association were additional elements of the inoculation process. I do not think he considered them competing processes or explanations.

2. Although memory was the most dominant factor across the studies, motivational threat was the second most dominant factor in several studies. Given that the motivational threat measure is still rather new, and Banas and Richards (2017) noted that it likely needed more refinement, would the authors have suggestions about how to improve the measure?

3. I'm not sure the affective "threat" booster was ever going to work. I get where you were trying to go with it, but I think that even if participants were going to be "threatened" or "apprehensive" or "motivated" then, theoretically, it would benefit them to have to content to process. That's the point motivation/threat component of the theory – it's the motivational catalyst to engage in the resistance process. Thinking through something, perhaps learning something new, and perhaps most importantly, remembering it.

4. I'd like to see the authors more directly address how memory would factor into different types of attacks, like the movie-based attacks used in Banas and Miller (2013). Additionally, how would memory account for cross-protection effects?

As you can see, I just have a couple suggestions for the discussion section. This perhaps the best inoculation study I've ever read. I'm impressed.

John Banas

Version 1:

Reviewer comments:

Reviewer #1

(Remarks to the Author)

I am happy with how the authors addressed all my comments.

Reviewer #2

(Remarks to the Author)

I would like to thank the authors for their careful responses to my comments. I am happy to recommend this version of the paper for publication, and would like to congratulate the authors on an impressive piece of work.

Reviewer #3

(Remarks to the Author)

I am satisfied with the revisions. I feel the authors made a sincere effort to address my concerns (as well as the concerns of other reviewers). I look forward to citing this study and sharing it with my students

Response to Reviewer Comments

Reviewer #1

The present manuscript describes five longitudinal experiments investigating the “long-term” effectiveness of psychological “inoculation”. The research addresses a long-needed aspect in the literature on inoculating against misinformation: A systematic assessment of its long-term effects. I applaud the authors for transparent reporting of results, including deviations from preregistration. However, I also have a couple of concerns, three major and several smaller ones, detailed below.

Major issues:

--The biggest point is this: Transparency and fairness in using the vaccination metaphor after the present results. Given the small effects of inoculation over time (e.g., day 30 in Figure 3), I wonder whether it is fair and accurate to still use the “inoculation” metaphor so broadly. One decisive feature of actual vaccinations is precisely their longevity, see the example of childhood vaccinations. This is clearly not the case here, or at least not under all the conditions where the metaphor is applied to. Given the partially small effect sizes after just one month, my honest opinion is this: I do not think it is fair to exploit that vaccination metaphor any longer. For the sake of transparency and honesty in reporting, especially to the general public, what the authors refer to as “inoculation” should simply be described as “providing information” or “gaming”, even if this means sounding less flashy. This is not only the case in order to fairly describe the long-term effects (or lack thereof) of these “inoculations” but also to make sure that the public does not come to believe that the effectiveness of actual vaccinations suffer from such dramatic decay rates, rather being (demonstrably!) effective for years or even decades.

=> First, thank you very much for the helpful and constructive feedback. We understand the concerns about using the term “inoculation”. However, we wish to stress that “inoculation theory” is a well-established term in psychology and the cognitive sciences that refers to a specific theory and associated procedures that have been used since the 1960s (McGuire, 1961, 1964) and several meta-analyses specifically examine the potential of “inoculative” communication procedures relative to *standard* education or information provision (Banas & Rains, 2010; Lu et al., 2023). It is therefore important to make this conceptual distinction clear as the metaphor refers to and is based on the procedure – which closely mimics that of a medical inoculation (i.e., exposure and refutation of a severely weakened dose of the misinformation rather than only factual information provision). Importantly, the analogy also refers to the *process* rather than the outcome of the inoculation procedure (Ivanov, 2017). We do not feel that it would be fitting for this paper to abandon this rich theoretical framework, as it would distract from the main contribution. And although some inoculation scholars have called inoculation interventions a “vaccine” against misinformation (McGuire, 1970), we do not use this language within the current paper. Meanwhile some inoculation scholars are arguing that we need to make *more* use of the analogy (Compton, 2020), as it could provide a valuable framework to inspire new methods of inoculation and expand the theory further.

However, we do recognize that the implication of “immunisation” could potentially be misinterpreted given modest effects that decay over time (as the reviewer notes) so to avoid any such misperceptions we now note in the discussion that psychological inoculation differs from biological immunisation in that it does not offer the same level of protection and wears off over time (p. 26).

We think this strikes a good compromise between maintaining important theoretical nuance whilst avoiding misperceptions about the ultimate outcome.

- Compton, J. (2020). Prophylactic versus therapeutic inoculation treatments for resistance to influence. *Communication Theory*, 30(3), 330–343. <https://doi.org/10.1093/ct/qtz004>
- Ivanov, B., & Parrott, R. (2017). Inoculation theory applied in health and risk messaging. In *The Oxford Encyclopedia of Health and Risk Message Design and Processing*. Oxford University Press. <https://doi.org/10.1093/acrefore/9780190228613.013.254>
- McGuire, W. J. (1970). Vaccine for brainwash. *Psychology Today*, 3(9), 36–39.
- McGuire, W. J. (1961). Resistance to persuasion conferred by active and passive prior refutation of the same and alternative counterarguments. *The Journal of Abnormal and Social Psychology*, 63(2), 326–332. <https://doi.org/10.1037/h0048344>

--Similarly, the title refers to “long-term” resistance. This is simply not fair, to put it mildly. No vaccination would be considered to be effective in the “long run” if it demonstrably works for 30 days. One month is surely longer than what is revealed in typical cross-sectional analyses, but it is surely not what counts as “long-term” with respect to vaccinations per se. Please tone down, even if only in the sake of not committing to the same errors as those you mean to inoculate against—after all, click-bait headlines only defeat your very purpose.

=> We appreciate the need for article titles to be accurate. Within the cognitive sciences, and especially in the memory literature, “long-term” is typically defined as lasting for longer than a few seconds. As an important part of this paper refers to mechanisms of memory, where anything longer than a few seconds is typically termed long-term memory, we argue that it is more consistent with this extensive literature to use this terminology. See the following article for an overview:

- <https://arc.duke.edu/how-long-short-term-memory-shorter-you-might-think>

We understand that people may not have the same understanding in all fields. In the social or even neuroscientific literature more generally, long-term benefits may be referred to as anything lasting for at least a couple of days. To give another example of this, here is a *Nature Communications* paper that explores the “long-term benefits” of an intervention by conducting an experiment that spans only 3 days:

- Gerlicher, A. M. V., Tüscher, O., & Kalisch, R. (2018). Dopamine-dependent prefrontal reactivations explain long-term benefit of fear extinction. *Nature Communications*, 9(1), 4294. <https://doi.org/10.1038/s41467-018-06785-y>

Taken together, in order to allow for comparisons with previous studies, we argue that any exploration of effects that go beyond one week should be called an exploration of long-term effectiveness in this context. In addition, the title of the paper indicates that the paper explores ways to *increase* the long-term effectiveness, keeping it open as to what the boundary conditions are for the maximum length of increasing this.

Furthermore, while we agree that biomedical vaccines typically last for much longer than 1 month, we would argue that this does not necessarily mean that any vaccine that one could possibly develop does so. For example, when looking at COVID-19 for example, there are people who are now invited to get their 7th dose of the vaccine, showing that the long-term effectiveness is not necessarily as long for every vaccine or every person. It is likely that *some* effects persist for inoculation interventions beyond 30 days even without any booster, but that the effect is decayed nevertheless, which is likely true for both psychological inoculation and for biomedical vaccines. However, it is very likely based on the memory-motivation model advanced in this paper that if someone would have more than 5 psychological inoculation boosters (as some have received for the COVID vaccine), the effects would last for many months and then even meet the most stringent criteria for long-term effectiveness.

--It would be preferable to report Bayesian statistics as opposed to frequentist results. This is for several reasons, among them providing more robust estimates and providing evidence for the Null.

=> We thank the reviewer for the suggestion to use Bayesian statistics as an alternative analysis. We have now added Bayesian statistics (Bayes Factor 10 + error %) for all tests in the main body of the paper. We also kept the frequentist statistics as we preregistered that we would report these (see our preregistration at https://aspredicted.org/blind.php?x=GPR_5FB). We found similar results using the Bayesian analyses so the good news is that all results are compatible (i.e., this is an excellent robustness confirmation).

Minor points:

--under: The Present Research: It is unclear what “these mechanisms” refers to

=> We have now clarified this by replacing “*these mechanisms*” with “*memory and motivation*” (p. 4).

--in how far constitutes the spreading of own misinformation in Bad News a “severely weakened form”?

=> We have now added between brackets “*i.e., using humorous examples that highlight the flaws in the misinformation in a safe, controlled environment*” (p. 5).

Thanks again for the valuable comments and feedback, which we hope to have addressed to the best of our ability given the stringent word limits.

Reviewer #2

This paper describes five experiments assessing long-term effectiveness of “inoculation” interventions for misinformation, and evaluating the role of memory for the intervention and motivational threat in the longevity of the effect. I appreciate the authors’ efforts to bring several related studies together to tell an overarching story about the effectiveness of inoculation. The five included studies are however very different from one another in rationale and methodology, and many important nuances are lost in the effort to find commonalities. Overall, I believe this paper can make a valuable contribution to the literature, but some clarifications are needed. I have listed a few additional points below.

1. I appreciate that this is not entirely within the authors’ control, but I found the structure of the paper very hard to follow. Methodological details are split across the Results section, the Methods section, the (very long) supplement and the As Predicted preregistrations. For other details, we are directed to previous papers (e.g. Maertens et al., (2020)). In order to understand exactly what was happening at any given point, I had to continually flick back and forth between all of these sources, and I am not confident that I have absorbed all the details correctly. Apologies if this leads to errors in any of my review points. To remediate this, I suggest including more methodological detail in the main text if possible; for example, information about the nature of the inoculation videos in studies 3-5 and the topic of misinformation in studies 2-5 is buried deep in the supplement, and I don’t believe the “false petitions” element is explained at all.

=> We thank the reviewer very much for the support and their excellent and constructive comments to make the paper clearer and flow better. We agree that some details can be added to aid understanding. We have added some further clarifications to the main text on the points mentioned: the false petition (p. 8), the topic of misinformation in studies 2–5 (p. 14; p. 18), and the nature of the inoculation videos (p. 18).

2. It is often difficult to establish exactly which effects are being discussed. For example, on page 9: “We first found that the inoculation effect was still significant at 8 days”. It is not clear what is meant by “inoculation effect” here. Is this simply a comparison between T1 and T10 within the inoculation group, or is it the difference between the inoculation and control group? These issues are present throughout the paper, but in most cases could be resolved by reporting the full ANOVA/ANCOVA results, rather than selected terms. These data might be best presented in table format.

=> We have added a clarification to highlight that all AN(C)OVA comparisons are comparisons between groups, and in this case between the inoculation group and the control group (p. 10). Due to the large range of different AN(C)OVA models used, and now with Bayesian analyses added as well (Reviewer 1), it would become too onerous and confusing to print *all* models (which would be more than 20 tables) in full in the paper instead of what we do now, which is focusing on the preregistered analyses. In addition, it would take us far over the word count and table count limits. However, for transparency, the tables can be reproduced in full by running the R scripts, and we have now also added them in ready-to-read format (R Markdown HTML printouts) on the OSF at <https://osf.io/mkswt> (Text

Paradigm), <https://osf.io/9qc8a> (Gamified Paradigm), and <https://osf.io/ersfc> (Video Paradigm).

3. I may be missing some important context here (I recognise that the authors are constrained by a tight word limit) but as written it seems that the control group is not being used as a comparator in the assessment of inoculation effects. Please clarify if this is the case, and if so, explain why. It seems like the interaction between group and time would be the most important term here as it would allow the authors to determine whether performance in the inoculation group(s) exceeds that in the control group at each time point.

=> Yes, thank you for this comment, please see our response to Question 2. We agree with your argument and we are glad to say that this is indeed what we did. The key inoculation effects as determined by the AN(C)OVA are determined by the comparison between the inoculation groups and the control groups at the different time points, as is now clarified (p. 10) and is also what we preregistered.

4. Study 1: The authors used two paired t-tests to separately examine the misinformation effect for the control and inoculation groups. They report a significant effect for the control group, but not for the inoculated group. I was surprised that the authors did not use a 2x2 ANOVA here as this approach does not allow inferences regarding the relative performance of the two groups (i.e. the fact that there was a misinformation effect in one group but not the other does not imply that the two groups differ significantly from each other). This comparison is embedded within the 3-way ANOVA, but this analysis is described so briefly that I had trouble following the interpretation.

=> We understand the confusion, apologies. We did in fact report a significant effect in the inoculation group as well, and have improved the wording now to make this clearer (p. 9; “*there was a positive effect on the perceived scientific consensus*”). We also agree that a relative performance analysis is interesting, and have now added this analysis as well (p. 10). We found the result (contrast: inoculation vs. control group posttest performance while controlling for pretest differences) to be significant with medium-to-large effects, $t(616) = 9.19$, $p_{\text{tukey}} < .001$, $d = 0.739$, 95% CI [0.576, 0.902], $\text{BF}_{10} = 1.069\text{e}+12$ (error < 0.001%).

5. Please include pre-test ratings of scientific consensus in Figure 3 for comparison purposes.

=> This is a great idea, thank you – we have now added it to Figure 3 (p. 13) and to Figure 4 (p. 17).

6. The motivational threat measure was not defined in the preregistration, and a number of different measures appear to have been selected here. Please provide further details about how the three items included in this measure were selected. The difference between “motivation threat” and “apprehensive threat” is not explained.

=> At the moment we are only aware of one measure of motivational threat in the inoculation literature, the one by (Banas & Richards, 2018), which we cite in our manuscript. The other scales that are included that you mention are exploratory. Nevertheless, we agree that it is useful to clarify and specify all of this in a bit more detail in the manuscript as readers may not be familiar with the different scales, so we have done this now on (p. 31) as suggested.

- Banas, J. A. & Richards, A. S. (2017). Apprehension or motivation to defend attitudes? Exploring the underlying threat mechanism in inoculation-induced resistance to persuasion. *Communication Monographs*, 84, 164–178.

7. Please provide details of the validity and reliability of the memory and threat scales. Were pilot data obtained for these measures?

=> Thanks, we have now clarified better in the paper how we chose the items for the scales, and added the Cronbach α for the memory and threat scales (p. 14, p. 19, pp. 30–32).

8. Study 2: Please remove references to trends in non-significant findings providing ‘mixed evidence’ for hypotheses or being “on the border of significance”. This also applies to Table S24 in the supplement, where clearly non-significant findings are listed as ‘mixed’. The analyses were well-powered but could not provide support for the hypotheses at the preregistered alpha level, so these claims are misleading.

=> That’s fair, we have now removed all statements linking non-significant changes to evidence for/against hypotheses throughout the paper, and have replaced the “mixed evidence” category by “no evidence” (frequentist) or “inconclusive” (Bayes when there was only anecdotal evidence for the null) for these.

9. Studies 3-5: Please clarify the primary outcome variables in the main text. This section refers to “discernment”, which implies the ability to discriminate between true and false news, but the method indicates that this is in fact “manipulativeness discernment”, which appears to measure whether or not participants can identify particular elements in a video. It is important not to conflate this with misinformation detection, since manipulative elements can be present in truthful information.

=> Yes there are different types of discernment, we have now clarified this better in the manuscript that we are concerned with helping people discern manipulation techniques (p. 18; “*manipulativeness discernment, whether people can distinguish manipulative social media posts from neutral ones*”).

We appreciate the reviewer is aware of the tight word limits so we have done our best to clarify all points to the best of our ability. Thanks again for the constructive and helpful feedback.

Reviewer #3

I have completed by review of NCOMMS-23-14656-T, entitled “Psychological Booster Shots Targeting Memory Increase Long-Term Resistance Against Misinformation.” The manuscript reports five pre registered longitudinal experiments that examine the effectiveness of various psychological inoculation interventions with nearly 12,000 participants in total. Overall, I am quite impressed with the totality of research in this manuscript. As with any study, I find some positive aspects especially noteworthy, and I have some suggestions for improvement.

Positives:

1. The topic of a misinformation is one of crucial importance, and I applaud the authors for researching this issue. As noted in the manuscript, discerning misinformation isn't just essential for adaptive decision making at individual and societal levels, the proliferation of misinformation has directly contributed to violence and death.
2. I think the authors make a strong case for messaging that is preemptive in nature, and I think studying inoculation theory in the context of addressing misinformation is very sensible.
3. The most impressive element of the manuscript for me is the testing of several different competing and complementary theoretical ideas across a number of studies. Studying the mechanisms of inoculation is a longstanding tradition in the literature, and looking at the roles of motivation and memory, especially as to how they may both mediate the inoculation process to affect misinformation discernment in the proposed memory-motivation model, is a valuable step forward for the inoculation literature.
4. In addition to the mechanisms, the focus on temporal dynamics of inoculation interventions also addressed other theoretical issues, like the related issues of message decay and booster shots. The role of message decay is of theoretical importance but also practical importance is misinformation interventions are actually to have a pronounced effect on the problem. I liked how the authors tied connected boosters and decay back to the research on memory.
5. In addition to the clever theorizing, the studies were carefully conducted and analyzed. And the Ns were impressive.

=> We are very thankful to the reviewer for the kind comments and thorough reading of our manuscript, and for encouraging us by listing the strong points next to the points for improvement.

Areas for improvement:

1. I believe Pfau et al. (2005) were not arguing against threat as much as arguing that memory and mental association were additional elements of the inoculation process. I do not think he considered them competing processes or explanations.

=> This is a good point and we have now made this clearer both in the main text (p. 4) and in the supplementary discussion (p. 75).

“Although Pfau et al. (2005) were not arguing against threat as much as arguing that memory and mental association were additional elements of the inoculation process (i.e., it was not considered by them to be competing processes or explanations), the evidence in this paper would suggest that although threat has an important role in the inoculation process, the memory processes are more essential (for longevity) than the threat processes.”

2. Although memory was the most dominant factor across the studies, motivational threat was the second most dominant factor in several studies. Given that the motivational threat measure is still rather new, and Banas and Richards (2017) noted that it likely needed more refinement, would the authors have suggestions about how to improve the measure?

=> This is a great question. In the revision we have added statistics on the internal reliability of the motivational threat scale to the paper, which showed that the measure is remarkably stable across the three studies (despite the slight adaptations), with Cronbach α values around .80–.90, indicating good-to-excellent consistency and higher stability than the memory measure.

When looking at item-level analysis, we found that the internal consistency of the adapted scale (note that this may not necessarily apply to the original scale) could be further improved by removing the first item of the three (“... motivates me to resist misinformation”). As a reminder, these were the items for each of the scales:

Study 1

Thinking about climate change misinformation ...
... motivates me to resist misinformation
... I feel ready to argue against manipulative headlines
... makes me want to defend my attitudes against deceptive news

Study 2

Thinking about online misinformation ...
... motivates me to resist misinformation
... I feel ready to argue against manipulative headlines
... makes me want to defend my attitudes against deceptive news

Studies 3–5

Thinking about emotionally manipulative language on social media ...
... motivates me to resist misinformation
... I feel ready to argue against emotional headlines
... makes me want to defend my attitudes against deceptive news

The high dominance of the measure, together with the high stability, indicates to us that the measure works very well already and needs little improvement. To us, the shift away from “fear-based” or “apprehensive” threat was a critical insight as in the psychological

inoculation process, motivation to counter-argue and resist misinformation appears much more important. As such, we hope the measure from Banas & Richards (2017) will become the new gold standard for assessing the role of motivation in inoculation.

3. I'm not sure the affective "threat" booster was ever going to work. I get where you were trying to go with it, but I think that even if participants were going to be "threatened" or "apprehensive" or "motivated" then, theoretically, it would benefit them to have to content to process. That's the point motivation/threat component of the theory – it's the motivational catalyst to engage in the resistance process. Thinking through something, perhaps learning something new, and perhaps most importantly, remembering it.

=> We agree with this important nuance and now discuss this in the manuscript (pp. 24–25).

"It could be argued that a standalone affective "threat" booster had little chance of working as even if participants were going to be threatened, apprehensive, or motivated, it would benefit them to have content to process as well. In other words, the motivation and threat components of inoculation theory point towards threat being the motivational catalyst to engage in the resistance process, and therefore thinking through something, and as the memory-motivation model would propose, to learn something new and remember it."

4. I'd like to see the authors more directly address how memory would factor into different types of attacks, like the movie-based attacks used in Banas and Miller (2013). Additionally, how would memory account for cross-protection effects?

=> We agree that it is worth addressing this and have now included a discussion of this (pp. 23–24).

"Given that this role for memory was found when the inoculation and test stimuli were the same and fact-based (Study 1), when they were different and technique or logic-based (Studies 2–5), and when the inoculation was a video but the test stimuli text (Studies 3–5), there is strong indication that the role of memory spans the scope of inoculation interventions and provides explanatory power across modalities. It also shows that people are able to flexibly apply remembered knowledge to new contexts and different stimuli—a concept known as cross-protection in inoculation theory⁴⁸—which is essential in a world of fast-changing misinformation."

"For text-based, gamified, and video-based interventions, we found that the effect shows a decay rate that is comparable to an exponential forgetting curve^{33,34}, and that the effect of specific text-based interventions can stay intact for about a month without a booster intervention, while the effects for the video-based and gamified interventions lost significance within the first two weeks without a booster intervention. This difference is likely due to the properties of specific issue-focused inoculations interventions, which are targeted to a limited amount of very specific content and may therefore be easier to remember and the attack stimuli easier to recognise—similar to the finding by Banas et al. that interventions based on specific facts were more effective than logic-based interventions⁴⁹. Meanwhile technique-focused inoculation interventions are broader and tap into multiple skills—as illustrated by

the results of Studies 3–5 that showed the inoculation provided cross-protection to a wide range of stimuli and techniques—but therefore seem to be either less memorable, or make it harder to recognise the attack stimuli (i.e., making it harder to activate the memory).”

48. Parker, K. A., Rains, S. A. & Ivanov, B. Examining the “blanket of protection” conferred by inoculation: The effects of inoculation messages on the cross-protection of related attitudes. *Communication Monographs* 83, 49–68 (2015).

49. Banas, J. A. & Miller, G. Inducing resistance to conspiracy theory propaganda: Testing inoculation and metainoculation strategies. *Human Communication Research* 39, 184–207 (2013).

As you can see, I just have a couple suggestions for the discussion section. This perhaps the best inoculation study I’ve ever read. I’m impressed.

=> We are extremely grateful for this meaningful comment. It really made our day, or even year. To hear this from a top inoculation scholar, gives us motivation and courage to continue this bold research programme. It will be forever in our memory.

We appreciate your extremely helpful feedback and have done our best to incorporate all of your points to the best of our ability given Nature’s word limits.